# Angiogenin mediates paternal inflammation-induced metabolic disorders in offspring through sperm tsRNAs

Yanwen Zhang[1], Li Ren[1], Xiaoxiao Sun[1], Zhilong Zhang[1], Jie Liu[1], Yining Xin[1], Jianmin Yu[2], Yimin Jia[1,3], Jinghao Sheng[2], Guo-fu Hu [4], Ruqian Zhao[1,3] & Bin He [1,3✉]

Paternal environmental inputs can influence various phenotypes in offspring, presenting tremendous implications for basic biology and public health and policy. However, which signals function as a nexus to transmit paternal environmental inputs to offspring remains unclear. Here we show that offspring of fathers with inflammation exhibit metabolic disorders including glucose intolerance and obesity. Deletion of a mouse tRNA RNase, Angiogenin (*Ang*), abolished paternal inflammation-induced metabolic disorders in offspring. Additionally, *Ang* deletion prevented the inflammation-induced alteration of 5′-tRNA-derived small RNAs (5′-tsRNAs) expression profile in sperm, which might be essential in composing a sperm RNA 'coding signature' that is needed for paternal epigenetic memory. Microinjection of sperm 30–40 nt RNA fractions (predominantly 5′-tsRNAs) from inflammatory *Ang*$^{+/+}$ males but not *Ang*$^{-/-}$ males resulted in metabolic disorders in the resultant offspring. Moreover, zygotic injection with synthetic 5′-tsRNAs which increased in inflammatory mouse sperm and decreased by *Ang* deletion partially resembled paternal inflammation-induced metabolic disorders in offspring. Together, our findings demonstrate that Ang-mediated biogenesis of 5′-tsRNAs in sperm contributes to paternal inflammation-induced metabolic disorders in offspring.

[1] Key Laboratory of Animal Physiology & Biochemistry, Ministry of Agriculture and Rural Affairs, College of Veterinary Medicine, Nanjing Agricultural University, 210095 Nanjing, PR China. [2] Institute of Environmental Medicine, Zhejiang University School of Medicine, 310058 Hangzhou, PR China. [3] MOE Joint International Research Laboratory of Animal Health & Food Safety, Nanjing Agricultural University, 210095 Nanjing, PR China. [4] Division of Hematology and Oncology, Department of Medicine, Tufts Medical Center, Boston 02111 MA, USA. ✉email: heb@njau.edu.cn

Increasing evidence indicates that paternal environmental inputs can affect the metabolic phenotype of the next generation through remodeling of the epigenetic blueprint of sperm. Recently, zygotic injection of total RNAs from isolated sperm or a subset of sperm small non-coding RNAs (sncRNAs) provided direct causal evidence for a role of a 'sperm RNA code' for the intergenerational transmission of paternally acquired phenotypes[1–9]. Sperm show dynamic changes in the composition of sncRNAs during the transition from the caput epididymis to the cauda epididymis, where sncRNAs are acquired via epididymosomes[4,10,11]. Thus, the epididymis response to environmental stimuli and the alteration of sncRNAs in the epididymis may play pivotal roles in shaping the profile of the sperm RNA signature and transmitting paternal environmental stressors to offspring. Inflammation is a complex physiological response of body tissues to stressful environmental stimuli, and is usually accompanied by alteration of sncRNAs[12–14]. The epididymis is exposed to a constant risk of inflammatory conditions resulting from bacterial infections, as well as idiopathic and noninfectious causes[15]. We and others have reported that the offspring from fathers with inflammation show alterations in growth, liver regeneration and immune responses in mice[16–18]. However, it is unclear whether the 'sperm RNA code' mediates paternal inflammation reprogramming of offspring phenotype.

Recently, studies have found that the 30–40 nucleotide (nt) sperm RNA fraction, predominantly consisting of tRNA-derived small RNAs (tsRNAs; also known as tRNA-derived fragments, tRFs), and its modifications can be rapidly altered by environmental inputs such as diet and exercise and contribute to intergenerational inheritance of acquired metabolic disorders[1–3,19,20]. Sperm tsRNAs can generate synergistic effects during early embryo development by regulating transcriptional cascades, transposon activities and other potential regulatory pathways that eventually affect offspring phenotypes[1,2,10]. Angiogenin (Ang) is a stress-responsive RNase (also known as RNase 5) that has been reported to mediate cleavage of mature tRNAs within the anticodon loops to produce 30–35 nt 5′-tsRNAs (also known as 5′-tRNA halves or tRNA-derived stress-induced RNAs, tiRNAs) and 40–50 nt 3′-tsRNAs that are linked with various stress responses[21–25]. Ang is also an acute phase reactant that increases during the inflammatory response[26–28]. The activation of Ang can facilitate tRNA fragmentation, leading to excessive amounts of tsRNAs and possibly eliciting pathophysiological conditions[29]. A recent study suggested that Ang is not the only RNase that can produce 5′-tRNAs[30]. In fact, Ang is not expressed at particularly high levels in the epididymis, and several epididymis-specific RNases are extremely highly expressed in this tissue[31,32]. Therefore, whether Ang-mediated tsRNAs biogenesis contributes to paternal inflammation effects on offspring has not yet been determined.

In this study, we used a mouse model to determine the effects of paternal inflammation on the metabolic health of adult offspring and explored the underlying mechanisms. We found that offspring of inflammatory fathers exhibit impaired glucose tolerance and elevated fat mass and Ang deletion abolished metabolic disorders in offspring. Our findings reveal that Ang-mediated 5′-tsRNAs biogenesis is required for the establishment of a sperm RNA 'coding signature' and for paternal inflammation-induced metabolic disorders in offspring.

## Results

**Paternal inflammation induces metabolic disorders in offspring.** Eight-week-old C57BL/6 male mice were injected with lipopolysaccharide (LPS) or saline once every other day for a total of four injections in 7 d to establish the inflammatory model, as previously described[17]. These mice were then allowed to mate with normal females for two days (Supplementary Tab. 1). Subsequently, offspring from control (Con) and inflammatory (Inf) dams were exposed to chow diet and water ad libitum without any challenge. The body weights of F1 male offspring were slightly, but significantly higher in male Inf F1 mice than in Con F1 mice from 8 weeks of age (Fig. 1a, b). Male Inf F1 mice maintained on the chow diet developed impaired glucose tolerance, showing significantly higher blood glucose levels during the glucose tolerance test (GTT) than Con F1 mice at 13 weeks of age (Fig. 1c). However, no significant differences were found in insulin tolerance test (ITT) and pyruvate tolerance test (PTT) results (Fig. 1d, e). Additionally, there were no statistically significant differences observed in blood insulin concentration (Supplementary Fig. 1a) and the content of hepatic glycogen between Inf F1 and Con F1 mice (Supplementary Fig. 1b). Notably, male Inf F1 mice displayed a significant elevation of perigonadal fat mass (Supplementary Fig. 1c) and fat-to-muscle ratio (Fig. 1f), accompanied by a 16% increase in the mean cross-sectional area of adipocyte cells (Fig. 1g, h, Supplementary Fig. 1d). The mass of gastrocnemius muscle was not significantly different between Inf F1 and Con F1 males, but the Inf F1 male mice exhibited lower exercise capacity as measured by maximum running speed in a treadmill test (Fig. 1i, Supplementary Fig. 1e). These results indicated that paternal inflammation offspring developed obesity and metabolic syndrome-like phenotypes.

Skeletal muscle is the principal site of glucose uptake and accounts for the regulation of glucose homeostasis[33,34]. To investigate whether alteration of glucose uptake in skeletal muscle contributed to glucose intolerance in Inf F1 mice, gastrocnemius muscles were sampled to measure the expression and distribution of glucose transporter type 4 (GLUT4), which is pivotal for glucose uptake in muscle tissue[35]. Western blot results revealed no significant difference in total GLUT4 content between Inf F1 and Con F1 mice (Supplementary Fig. 1f). Glucose uptake in skeletal muscle mainly depends on GLUT4 translocation from intracellular storage depots to the plasma membrane[36]. Notably, male Inf F1 mice displayed a significant decrease of GLUT4 content in membrane proteins of muscle (Supplementary Fig. 1f). Immunohistochemical analysis showed that GLUT4 is expressed in the muscle cell membrane in male Con F1 mice (Fig. 1j). In contrast, the amount of GLUT4 in muscle surface membrane was markedly lower and GLUT4-positive vesicles were detected in the cytoplasm in male Inf F1 mice (Fig. 1k). To obtain a global view of the transcriptional response, RNA-seq was carried out on RNA purified from muscle samples from either Inf F1 or Con F1 mice. As expected, RNA-seq analysis revealed that differentially expressed genes are dominantly enriched in metabolic pathways (including type II diabetes mellitus, the insulin signaling pathway, and the JAK-STAT signaling pathway) (Fig. 1l, Supplementary Fig. 1g–i).

**Ang is essential for paternal inflammation-induced metabolic disorders in offspring.** Next, we investigated the expression levels of Ang in the testis, caput epididymis, and cauda epididymis where sperm develop and undergo maturation. After LPS treatment for 24 h, there was an upregulation in the expression of Ang in the caput epididymis but not testis and cauda epididymis (Fig. 2a). The protein level of Ang was also upregulated in the caput epididymis (Fig. 2b). The lack of significant difference in the production of proinflammatory (IL1β, TNFα and IL6) and anti-inflammatory (IL10) cytokines in serum and caput epididymis between LPS-treated $Ang^{+/+}$ and $Ang^{-/-}$ mice indicated that Ang deletion did not protect mice from LPS-induced inflammation in caput epididymis (Supplementary Fig. 2a–l).

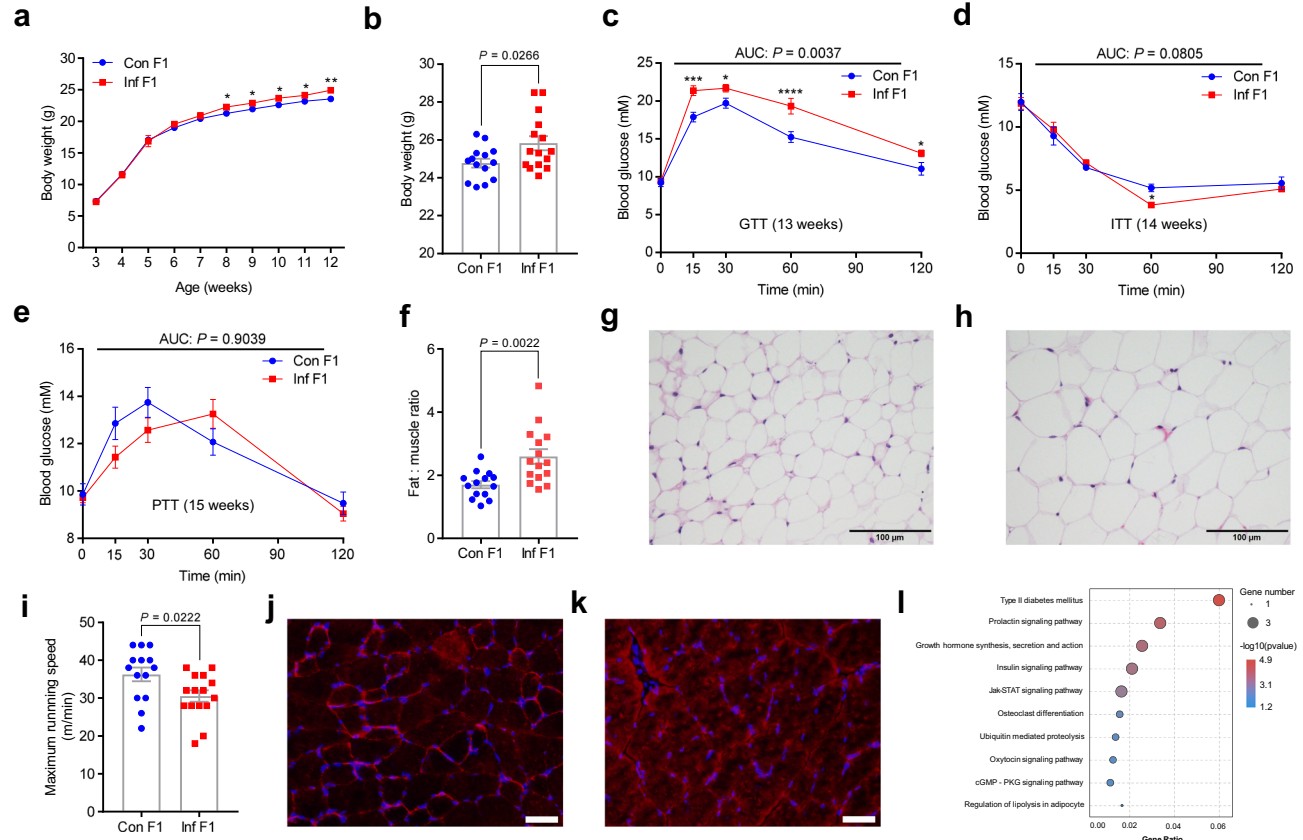

**Fig. 1 Body weight and metabolic parameters of F1 males with paternal inflammation.** Male mice were treated with LPS (Inf) or saline (Con) once every other day for a total of four injections in 7 days to establish an inflammatory model. These mice were then allowed to mate with normal females for two days. **a** Growth curves for F1 male offspring from Con- and Inf- fathers. Statistical analysis was performed by a MIXED procedure for repeated measures. *$P < 0.05$; **$P < 0.01$. **b** Body weight of male offspring at 15 weeks of age. **c–e** Blood glucose by GTT at 13 weeks (**c**), ITT at 14 weeks (**d**), and PTT at 15 weeks (**e**) of age. In (**c**, **d**) and (**e**), $n = 7$ mice per group. AUC, area under the curve. *$P < 0.05$; ***$P < 0.001$; ****$P < 0.0001$. **f** Ratio of fat-to-muscle weight of male offspring at 15 weeks of age. **g, h** Representative images showing adipocyte area in perigonadal fat of Con F1 (**g**) and Inf F1 (**h**) male mice. $n = 5$ mice per group. Scale bar = 100 μm. **i** Treadmill test results of male offspring at 12 weeks of age. In (**b**, **f**) and (**i**), $n = 14$ in Con and $n = 15$ in Inf. Statistical analysis was performed by two-tailed unpaired Student's $t$ test (**b**, **f**, **i**), by two-tailed, two-way analysis of variance (ANOVA), uncorrected Fisher's least significant difference (LSD) (**c–e**). Data are show as mean ± SEM. **j, k** Representative images showing cryosections of gastrocnemius muscle (cross sections) from Con F1 (**j**) and Inf F1 (**k**) mice stained with antibodies against GLUT4 (red) and DAPI (blue). $n = 5$ mice per group. Scale bar = 50 μm. **l** The top ten significant clusters of gene ontology (KEGG) terms enriched in gastrocnemius muscle samples of Con F1 and Inf F1 male offspring determined by GSEA and clustered under parent terms were related to synaptic signaling ($n = 4$, NES > |1.5| and FDR < 0.05). Source data are provided as a Source Data file.

To investigate the potential role of Ang in paternal inflammation-induced metabolic disorders, we used the $Ang^{+/+}$ Con, $Ang^{+/+}$ Inf, $Ang^{-/-}$ Con, and $Ang^{-/-}$ Inf models, as generated above, mated these mice with normal females, and then examined the metabolic parameters of the resulting progeny (Fig. 2c, Supplementary Tab. 2). The body weight of $Ang^{+/+}$ Inf F1 male mice showed a slight, but significant increase compared to those of $Ang^{+/+}$ Con (Fig. 2d). Moreover, $Ang^{+/+}$ Inf F1 displayed an elevation of perigonadal fat mass (Supplementary Fig. 3a) and fat-to-muscle ratio (Fig. 2e), accompanied by an increase in the mean cross-sectional area of adipocyte cells (Supplementary Fig. 3b, c). Furthermore, $Ang^{+/+}$ Inf F1 males showed significantly higher glucose than the other F1 males as assayed by GTT (Fig. 2f, g), but not ITT (Supplementary Fig. 3d, e). However, $Ang^{-/-}$ Inf F1 males exhibited no significant difference in glucose tolerance, similar to what was observed for $Ang^{+/+}$ Con F1 and $Ang^{-/-}$ Con F1 mice (Fig. 2f, g). There was no significant difference in gastrocnemius muscle mass (Supplementary Fig. 3f), but $Ang^{+/+}$ Inf F1 mice exhibited lower treadmill running capacity as measured by maximum running speed in a treadmill test (Fig. 2h). However,

male $Ang^{-/-}$ Inf F1 mice did not exhibit any obvious metabolic disorders or impaired treadmill running capacity, similar to what was observed for $Ang^{+/+}$ Con F1 and $Ang^{-/-}$ Con F1 mice (Fig. 2e–h, Supplementary Fig. 3a–f). The expression level of SOCS3 but not SOCS1, SOCS2 and ISR1 was higher in $Ang^{+/+}$ Inf F1 males compared with other F1 males (Supplementary Fig. 3g–j).

**Inflammation and Ang mutation alter cauda sperm tsRNAs profile.** To determine whether the tsRNA composition in sperm can be affected by inflammation and Ang deletion, we performed small RNA deep sequencing on cauda sperm isolated from $Ang^{+/+}$ Con, $Ang^{+/+}$ Inf, $Ang^{-/-}$ Con, and $Ang^{-/-}$ Inf mice, as generated above. The tsRNAs can be classified into four groups according to the region of tRNAs from which they are derived: 5′-tsRNAs, 3′-tsRNAs, CCA-tsRNAs, and other tsRNAs. Among the four types of tsRNAs, the 5′-tsRNAs were detected as the most abundant subtype in cauda epididymal sperm (ranging from 72.4% to 80.2%), and 3′-tsRNAs, CCA-tsRNAs, and other tsRNAs were detected as smaller fractions (Fig. 3a–h). Interestingly, both inflammation and Ang deletion

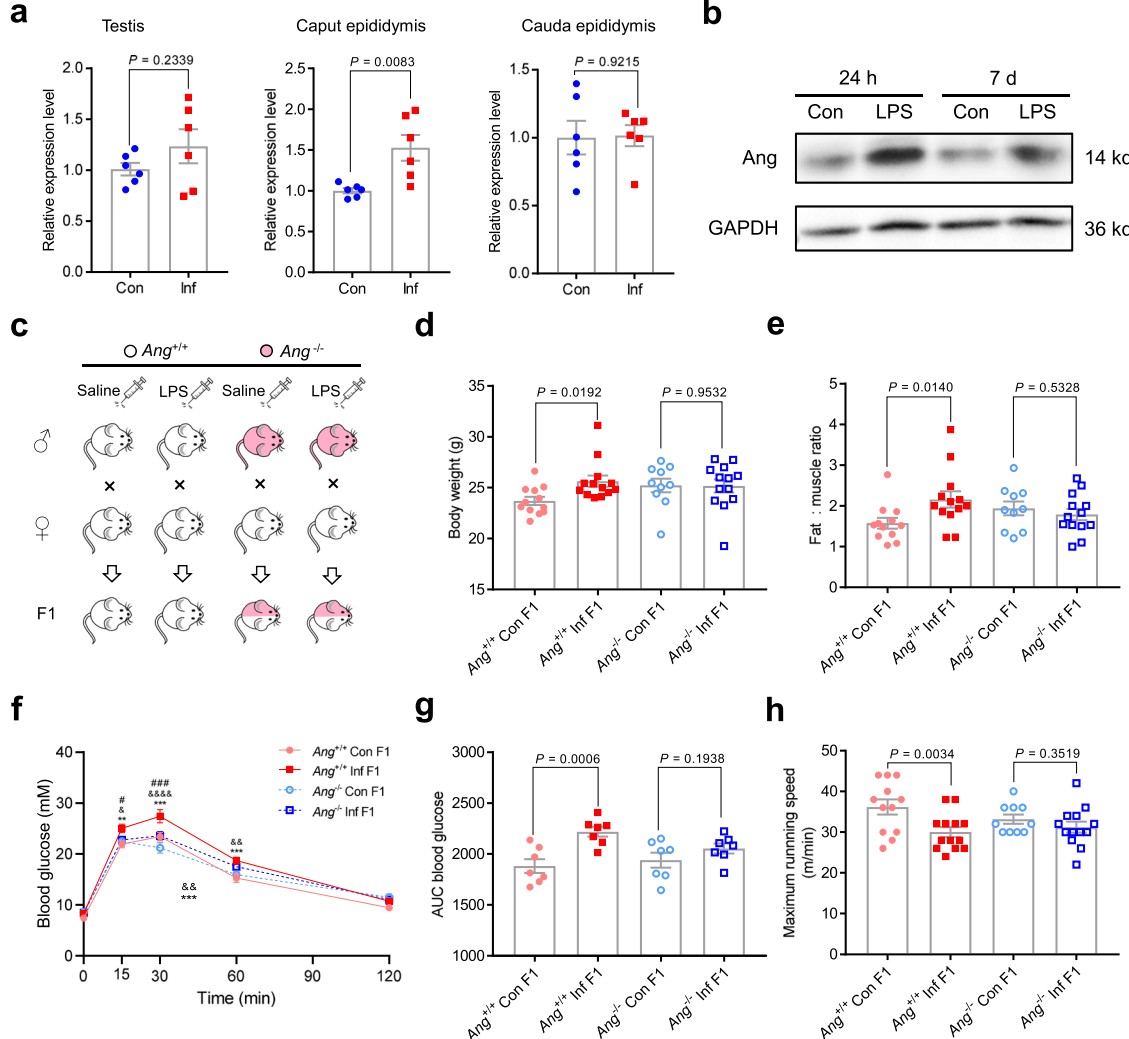

**Fig. 2 Expression of Ang in inflammatory mice and metabolic parameters of F1 males from inflammatory and *Ang*-deleted mice. a** The relative expression level of *Ang* mRNA in mouse testis, caput epididymis, and cauda epididymis under LPS (Inf) or saline (Con) treatment for 24 h. **b** Western blots of Ang protein in mouse caput epididymis under LPS or saline treatment for 24 h or 7 d. In (**a**, **b**), $n = 6$ mice per group. **c** The $Ang^{+/+}$ and $Ang^{-/-}$ F0 male mice were injected with LPS or saline once every other day for a total of four injections in 7 d in order to establish an inflammatory model. These mice were then allowed to mate with normal females for two days. Subsequently, offspring mice were exposed to chow diet and water ad libitum without any challenge throughout their lives. **d** Body weight of F1 males at 16 weeks of age. **e** Ratio of fat-to-muscle weight of F1 males at 16 weeks of age. **f** Blood glucose levels as assayed by GTT of F1 male mice at 14 weeks of age. ** $P < 0.01$, *** $P < 0.001$ ($Ang^{+/+}$ Inf versus $Ang^{+/+}$ Con); $^{\&}P < 0.05$, $^{\&\&}P < 0.01$, $^{\&\&\&\&}P < 0.0001$ ($Ang^{+/+}$ Inf versus $Ang^{-/-}$Con); $^{\#}P < 0.05$, $^{\#\#\#}P < 0.001$ ($Ang^{+/+}$ Inf versus $Ang^{-/-}$ Inf). **g** Area under the curve (AUC) statistics for (**f**). In (**f**, **g**), $n = 7$ mice per group. **h** Treadmill test results of F1 males at 8 weeks of age. In (**d**, **e**) and **h**, $n = 12$ in $Ang^{+/+}$ Con group, $n = 13$ in $Ang^{+/+}$ Inf group, $n = 10$ in $Ang^{-/-}$ Con group, and $n = 13$ in $Ang^{-/-}$ Inf group. Statistical analysis was performed by two-tailed unpaired Student's *t* test (**a**), by two-tailed, one-way ANOVA (**d**, **e**, **g**, **h**) or two-way ANOVA (**f**), uncorrected Fisher's LSD. Data are show as mean ± SEM. Source data are provided as a Source Data file.

affected the expression of individual sperm 5′-tsRNAs. Inflammation induced a slight increase in the proportion of 5′-tsRNAs and *Ang* deletion led to an obvious decrease in the proportion of 5′-tsRNAs (Fig. 3a–h). Because 30–35 nt 5′-tsRNAs are extremely enriched in mature mouse sperm[37], we analyzed these RNAs in further detail. Among the 30–35 nt 5′-tsRNA, expression levels of 88 5′-tsRNAs were increased and those of 17 genes were decreased in $Ang^{+/+}$ Inf compared with $Ang^{+/+}$ Con mice (Fig. 3i, Supplementary Data 1). *Ang* deletion almost completely negated the effect of inflammation on the sperm 5′-tsRNAs that originate from both nucleus- and mitochondria-encoded tRNAs (Fig. 3i–m, Supplementary Data 1). Prominent examples of inflammation-induced tsRNAs include 5′-tsRNA-Gly-GCC, 5′-tsRNA-iMet-CAT, and 5′-mt-tsRNA-Val-TAC, whose abundance displayed ~two- to threefold increases in

inflammatory mouse sperm and decreased by *Ang* deletion to baseline levels (Fig. 3j–m, Supplementary Fig. 4, Supplementary Data 1). Strikingly, several 5′-tsRNAs such as 5′-tsRNA-Cys-GCA, were upregulated in inflammatory mice but expression levels were not reversed by *Ang* deletion (Fig. 3n, Supplementary Fig. 4, Supplementary Data 1). The change of 5′-tsRNA-Gly-GCC level was also confirmed by Northern blot analysis (Fig. 3o). These data revealed that inflammation-induced pronounced changes in sperm tsRNAs composition and that *Ang* deletion can block the effects of inflammation on the sperm 5′-tsRNA profile.

Previous work suggested that sncRNAs are first synthesized in the epididymis and then trafficked to maturing sperm via epididymosomes[4,10,11]. To determine whether the changes of tsRNAs in cauda sperm are similar to those in the caput epididymis in

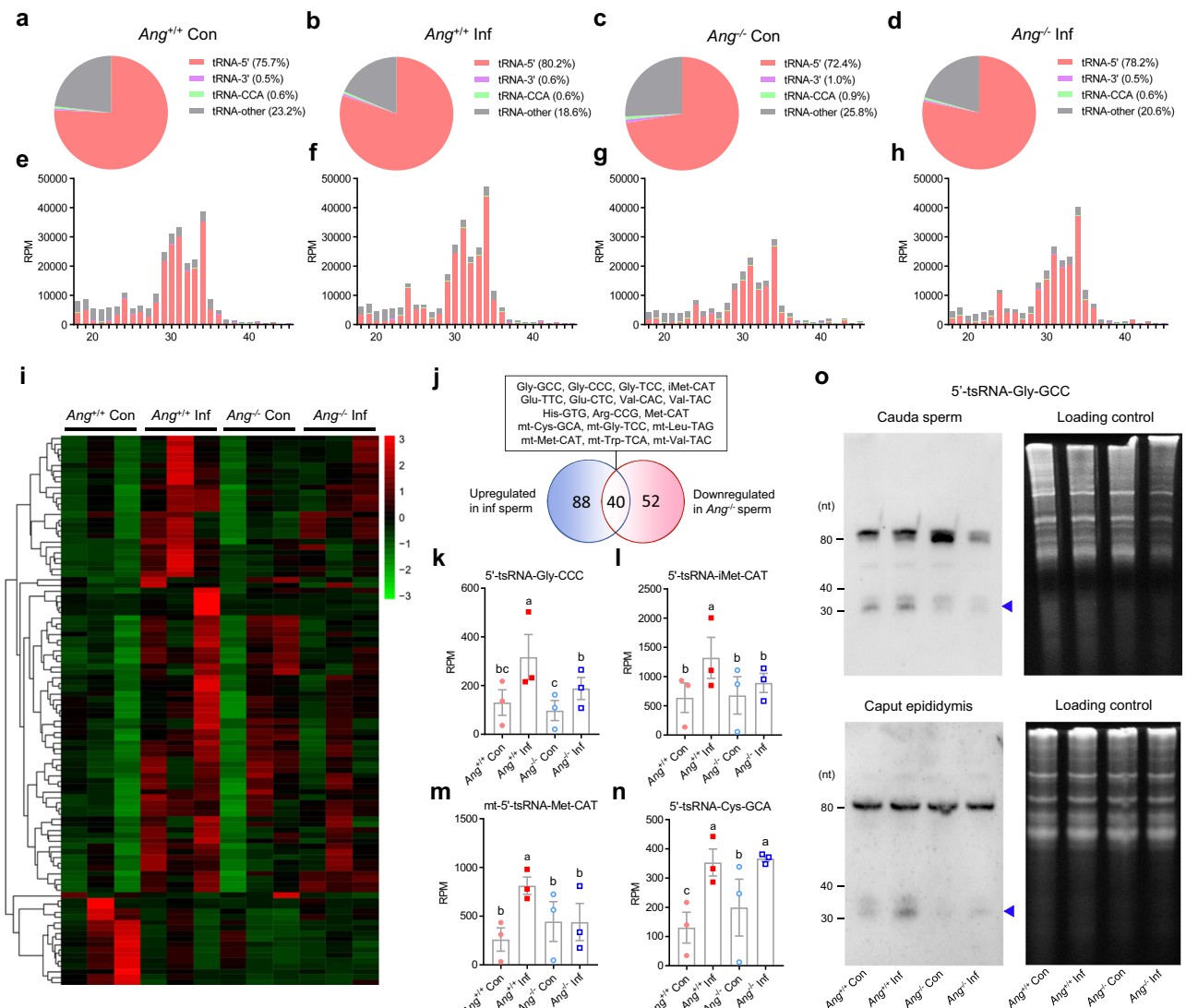

**Fig. 3 Sperm tsRNAs profile in inflammatory and *Ang*-deleted mice.** Male *Ang*$^{+/+}$ and *Ang*$^{-/-}$ mice were treated with LPS (Inf) or saline (Con) once every other day for a total of four injections in 7 d to establish an inflammatory model. Small RNA deep sequencing was performed on cauda sperm isolated from *Ang*$^{+/+}$ Con, *Ang*$^{+/+}$ Inf, *Ang*$^{-/-}$ Con, and *Ang*$^{-/-}$ Inf mice. Sperm total RNAs extracted from two mice were mixed together for Small RNA deep sequencing. **a–d** Subcellular fractionation of tsRNAs (5′-tsRNAs, 3′-tsRNAs, CCA-tsRNAs, and other tsRNAs) in *Ang*$^{+/+}$ Con (**a**), *Ang* $^{+/+}$ Inf (**b**), *Ang*$^{-/-}$ Con (**c**), and *Ang*$^{-/-}$ Inf (**d**) sperm. **e–h** Length distribution and pattern changes of different tsRNAs in *Ang*$^{+/+}$ Con (**e**), *Ang*$^{+/+}$ Inf (**f**), *Ang*$^{-/-}$ Con (**g**), and *Ang*$^{-/-}$ Inf (**h**) sperm. **i** Heat map of differentially expressed 30–35 nt 5′-tsRNAs in sperm. **j** Venn diagram showing numbers of 5′-tsRNAs with significant upregulated in sperm from inflammatory mice and downregulated in sperm from Ang-deleted mice. **k–n** Expression levels of select 5′-tRNAs that are altered with inflammation and *Ang* deletion. Statistical analysis was performed by two-tailed, one-way ANOVA, uncorrected Fisher's LSD. $n = 6$ mice per group. All data are plotted as mean ± SEM. Values with different superscripts are significantly different from each other ($P < 0.05$). **o** Northern blot analyses of 5′tsRNA-Gly (shown by arrow heads) in *Ang*$^{+/+}$ Con, *Ang* $^{+/+}$ Inf, *Ang*$^{-/-}$ Con, and *Ang*$^{-/-}$ Inf cauda sperm and caput epididymis. $n = 6$ mice per group. Sperm total RNAs extracted from two mice were mixed together for each lane in the experiment. Sperm total RNAs were run on a 15% denatured PAGE gel as shown as a loading control. Blots are shown as representatives of three independent experiments with similar results. Source data are provided as a Source Data file.

inflammatory and *Ang*$^{-/-}$ model, we detected the abundances of the most highly expressed tsRNAs in cauda sperm including 5′-tsRNA-Gly, 5′-tsRNA-Glu, 5′-tsRNA-Val and 5′-tsRNA-Cys in the caput epididymis by Northern blot. The results showed that the abundances of 5′-tsRNA-Gly, 5′-tsRNA-Glu and 5′-tsRNA-Val were increased in the inflammatory caput epididymis and decreased by *Ang* deletion (Fig. 3o, Supplementary Fig. 5). This was consistent with sperm small RNA-seq analysis showing higher levels of 5′-tsRNA-Gly, 5′-tsRNA-Glu and 5′-tsRNA-Val in cauda sperm of *Ang*$^{+/+}$ Inf mice compared with *Ang*$^{+/+}$ Con, *Ang*$^{-/-}$ Inf and *Ang*$^{-/-}$ Con males. Differently, the abundance of 5′-tsRNA-Cys was upregulated in inflammatory caput

epididymis but was not reversed by *Ang* deletion (Supplementary Fig. 5). Interestingly, these results were consistent with the previous finding that the biogenesis of 5′-tsRNA-Gly, 5′-tsRNA-Glu and 5′-tsRNA-Val are Ang-dependent but that of 5′-tsRNA-Cys is Ang-independent[30].

**Sperm tsRNAs of *Ang* deletion inflammatory males rarely induce metabolic disorders in offspring.** To assess whether the sperm tsRNAs can pass paternal information to their offspring, we used the *Ang*$^{+/+}$ Con, *Ang*$^{+/+}$ Inf, *Ang*$^{-/-}$ Con, and *Ang*$^{-/-}$ Inf

models as generated above, and a previously established zygotic sperm RNA injection protocol (Fig. 4a)[1,2]. We collected the 30–40 nt sperm RNA fractions from $Ang^{+/+}$ Con, $Ang^{+/+}$ Inf, $Ang^{-/-}$ Con, and $Ang^{-/-}$ Inf males, and performed zygotic injections, using injection of water as a control (because the RNAs injected in the other four groups were dissolved in water). We then examined the body weight, fat mass, muscle mass, and metabolic parameters of the F1 male offspring at 17 weeks of age. RNA injection did not adversely affect embryo development compared with control injection (Supplementary Tab. 3). The F1 male offspring of the $Ang^{+/+}$ Inf group showed impaired glucose tolerance, as evidenced by significantly higher glucose levels assayed by GTT compared with the levels in $Ang^{+/+}$ Con and $Ang^{-/-}$ Con males, as well as the control injection group (Fig. 4b, c, Supplementary Fig. 6a, b). In contrast, the F1 male offspring from the $Ang^{-/-}$ Inf group did not show metabolic disorders relative to the $Ang^{+/+}$ Inf group, but instead exhibited a similar pattern to that of $Ang^{+/+}$ Con and $Ang^{-/-}$ Con males and the control injection group (Fig. 4b, c). The body weights of $Ang^{+/+}$ Inf F1 male mice showed a slight, but significant increase compared to those of $Ang^{-/-}$ Con and control injection F1 mice (Fig. 4d). Consistent with these findings, $Ang^{+/+}$ Inf offspring also displayed higher fat-to-muscle mass ratio compared with the levels in $Ang^{+/+}$ Con, $Ang^{-/-}$ Inf, and $Ang^{-/-}$ Con males, as well as the control injection group (Fig. 4e, Supplementary Fig. 6c–e). Moreover, $Ang^{+/+}$ Inf offspring exhibited lower treadmill running capacity compared with the level in the control injection group (Supplementary Fig. 6f). The expression level of SOCS3 was higher in $Ang^{+/+}$ Inf F1 males than $Ang^{-/-}$ Con F1 males (Supplementary Fig. 6g–j). Together, the results from zygotic injection of 30–40 nt RNA fractions strongly suggest that Ang deletion in inflammatory males abolished the ability of sperm RNAs to induce metabolic phenotypes in their offspring.

We next synthesized a combination of the most highly expressed 5′-tsRNAs whose abundances displayed increases in inflammatory mouse sperm and decreased by Ang deletion to baseline levels in the sperm (Fig. 3j, Supplementary Fig. 4, Supplementary Tab. 4) for zygotic injection to investigate whether they could mimic the function of endogenous sperm tsRNAs. Scrambled RNA and sperm 30–40 nt RNAs from $Ang^{+/+}$ Inf mice as generated above (sperm RNAs) and male mice obtained from natural mating were used as a control (Supplementary Tab. 5). The F1 male offspring from the sperm RNAs group showed impaired glucose tolerance, as evidenced by significantly higher glucose levels assayed by GTT compared with the levels in scrambled RNA and synthetic tsRNAs group, as well as the control group (Fig. 4f, g, Supplementary Fig. 7a, b). Strikingly, the F1 males of synthetic tsRNAs group showed a slight, but significant increase of glucose levels compared to those of the scrambled RNA group (Fig. 4f, g). Moreover, the resultant offspring from the synthetic tsRNAs injection group displayed higher body weight compared with the levels in scrambled RNA, sperm RNAs, and control groups (Fig. 4h). Consistent with these findings, mice in the synthetic tsRNAs and sperm RNAs group displayed an elevation of perigonadal fat mass (Supplementary Fig. 7c) and fat-to-muscle ratio (Fig. 4i), accompanied by an increase in the mean cross-sectional area of adipocyte cells (Supplementary Fig. 7d). There was no significant difference in gastrocnemius muscle mass (Supplementary Fig. 7e), but the synthetic tsRNAs injection group exhibited lower treadmill running capacity compared with the levels in scrambled RNA, sperm RNAs groups, as well as the control group (Supplementary Fig. 7f). The expression level of SOCS3 was higher in the sperm RNAs group than the scrambled RNA and the control group while the expression level of SOCS2 were higher in the synthetic tsRNAs group than the control group (Supplementary Fig. 7g–j). Together, the results showed that zygotic injection of a pull of synthetic tsRNAs can partly induce metabolic phenotypes in offspring.

## Discussion

The possibility that paternal preconceptional environmental inputs can influence various phenotypes in offspring has tremendous implications for basic biology as well as public health and policy[1,2,38–44]. Here, we described a mouse model in which paternal inflammation led to metabolic disorders including glucose intolerance and obesity in offspring. Metabolic disorders in offspring were not observed when $Ang^{-/-}$ father mice were used, suggesting that Ang participates in inflammation-induced metabolic disorders in offspring. Strikingly, Ang deletion prevented the alteration of the 5′-tsRNAs expression profiles in sperm that were induced by inflammation. Zygotic injection of sperm 30–40 nt RNA fractions (predominantly 5′-tsRNAs) from $Ang^{+/+}$ Inf but not $Ang^{-/-}$ Inf mice induced metabolic changes in the resultant offspring. Moreover, zygotic injection with synthetic 5′-tsRNAs which increased in inflammatory mouse sperm and decreased by Ang deletion partially resembled paternal inflammation-induced metabolic disorders in offspring. These converging clues suggest that Ang-mediated biogenesis of 5′-tsRNAs in sperm contributes to paternal inflammation-induced metabolic disorders in offspring.

There are several specific RNases that can cleave tRNAs at the anticodon loop for fragmentation into tsRNAs[30,45]. Here, we observed that Ang upregulation in the caput epididymis of inflammatory mice coincides with an increased level of several high-abundance tsRNAs including 5′-tsRNA-Gly, 5′-tsRNA-Glu, and 5′-tsRNA-Val. Importantly, the increases of these tsRNAs were reversed by Ang deletion. Previous work found that sncRNAs are first synthesized in the caput epididymis and then trafficked to maturing sperm via epididymosomes[4,10,11]. This requirement for Ang was further supported by comparison of 5′-tsRNA profiles in cauda sperm of $Ang^{+/+}$ Inf mice with those of $Ang^{+/+}$ Con, $Ang^{-/-}$ Inf, and $Ang^{-/-}$ Con males. Building on the recent work characterizing sperm RNA-mediated transmission of paternally acquired traits to offspring[1,2,4,5], our work identified Ang as a genetic factor that is essential in shaping the sperm tsRNAs profile that is responsible for paternal inflammation-induced metabolic disorders in offspring. Ang is upregulated in response to various environmental factors, including lack of amino acids, heat shock, ultraviolet radiation, hypoxia, oxidative stress, osmotic stress, and pathogen exposure, and this upregulation mediates an increase in tsRNAs[23,26,46]. A similar mechanism may program phenotypes in offspring in response to paternal environmental factors that induce Ang upregulation in epididymis. This mechanism could be highly relevant to human health because Ang function is highly conserved and has evolved to respond to many diverse environmental challenges[47].

Increasing evidence now suggests that sperm tsRNAs can mediate intergenerational effects but the underlying processes and mechanisms remain puzzling. Transfection of tsRNAs promoted lineage differentiation in embryoid bodies and embryonic stem cells in mice[48,49]. Furthermore, transfection of a pool of tsRNAs but not single tsRNA, downregulated the mitochondria oxidative phosphorylation and translation/ribosome pathways[48]. The induction of a biased metabolic state in the early embryo might alter metabolites or mitochondrial function to further trigger a chain reaction that continuously influences the metabolic state[50]. Thus, the phenotypic outcome of injecting 30–40 nt RNA fractions or a pool of synthetic tsRNAs could be a combinatorial effect and may relate to the function of tsRNAs in cell fate regulation in the early embryo. Chen et al. found that RNA modifications are essential for tsRNAs to exert their intergenerational effects, as non-modified synthetic 5′-tsRNAs did not induce glucose intolerant and insulin resistant in the offspring[1]. In line with their findings, zygotic injection with synthetic 5′-tsRNAs only partially resembled paternal inflammation-induced

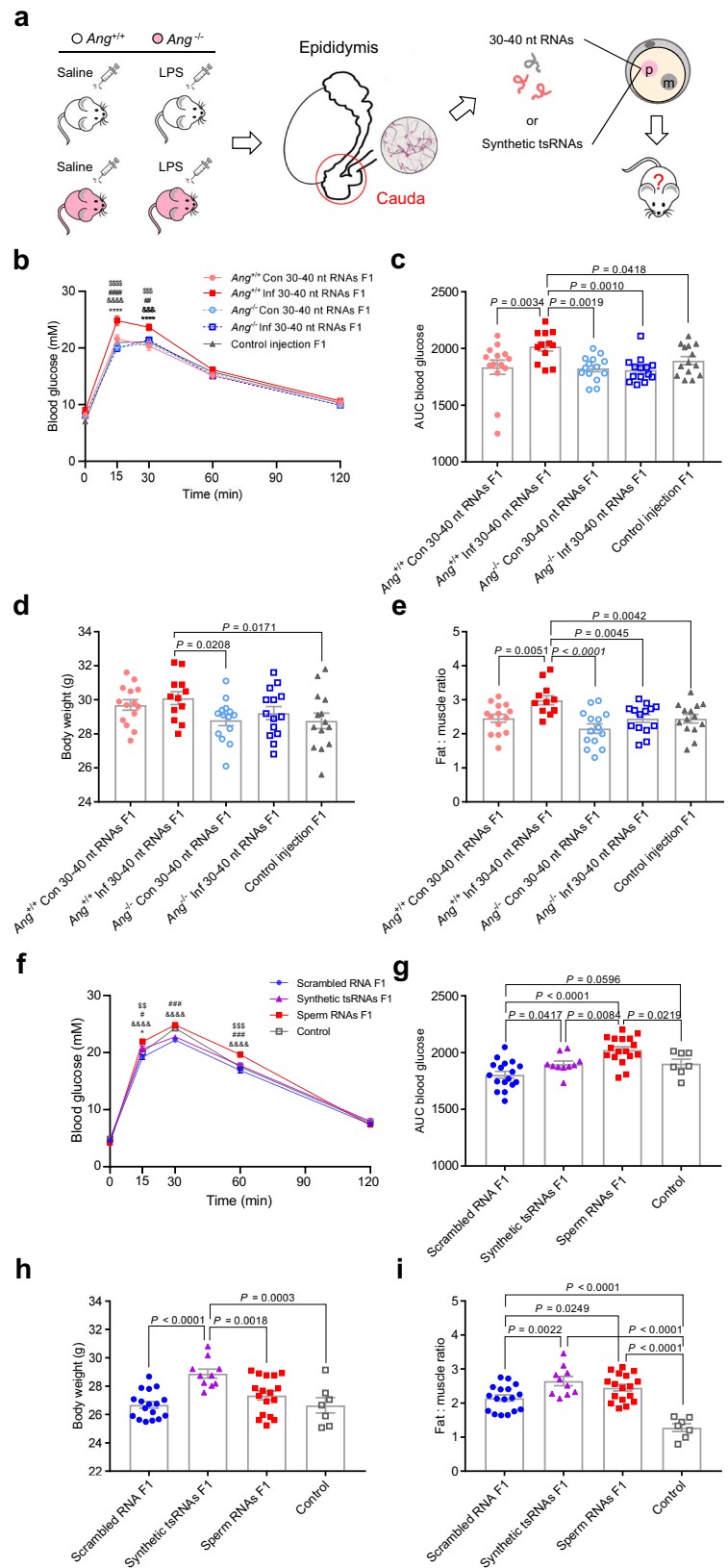

metabolic disorders in offspring in the present study. The molecular mechanism underlying 5′-tsRNAs actions appears to be more complex. In the present study, non-modified synthetic 5′-tsRNAs induced significantly less glucose intolerant yet similar obesity compared to the sperm RNAs. It is speculated that 5′-tsRNAs may exert their actions via RNA modifications-dependent or independent manners. Further work is needed to elucidate the function of sperm epitranscriptome in association with tsRNAs-mediated intergenerational effects. It is noted that mice from zygotic scrambled RNAs injection had higher fat mass

**Fig. 4 Body weight and metabolic parameters of F1 males generated by zygotic injection of sperm 30–40 nt RNAs or synthetic tsRNAs. a** Illustration of zygotic injection of sperm 30–40 nt RNAs or a pool of synthetic tsRNAs to generate F1 male offspring for phenotypic examination. **b** Blood glucose during GTT of F1 males generated from sperm 30–40 nt RNAs injection at 15 weeks of age. **** $P < 0.0001$ ($Ang^{+/+}$ Inf versus $Ang^{+/+}$ Con); $^{\&\&\&}$ $P < 0.001$, $^{\&\&\&\&}$ $P < 0.0001$ ($Ang^{+/+}$ Inf versus $Ang^{-/-}$ Con); $^{\#\#}$ $P < 0.01$, $^{\#\#\#\#}$ $P < 0.0001$ ($Ang^{+/+}$ Inf versus $Ang^{-/-}$ Inf); $^{\$\$\$}$ $P < 0.001$, $^{\$\$\$\$}$ $P < 0.0001$ ($Ang^{+/+}$ Inf versus control injection). **c** Area under the curve (AUC) statistics for (**b**). **d** Body weight of F1 males at 17 weeks of age. **e** Ratio of fat-to-muscle weight of F1 males at 17 weeks of age. In (**b–e**), $n = 14$ in $Ang^{+/+}$ Con group, $n = 12$ in $Ang^{+/+}$ Inf group, $n = 14$ in $Ang^{-/-}$ Con group, $n = 14$ in $Ang^{-/-}$ Inf group, and $n = 14$ in control group. **f** Blood glucose during GTT of F1 males generated from scrambled RNA, synthetic tsRNAs (synthetic tsRNAs), and sperm 30–40 nt RNAs from $Ang^{+/+}$ Inf mice (sperm RNAs) injection and normal male mice (control) at 12 weeks of age. * $P < 0.05$ (scrambled RNA versus synthetic tsRNAs); $^{\&\&\&\&}$ $P < 0.0001$ (scrambled RNA versus sperm RNAs); $^{\#}$ $P < 0.05$, $^{\#\#\#}$ $P < 0.001$ (synthetic tsRNAs versus sperm RNAs); $^{\$\$}$ $P < 0.01$, $^{\$\$\$}$ $P < 0.001$ (sperm RNAs versus control). **g** AUC statistics for (**f**). **h** Body weight of F1 males at 15 weeks of age. **i** Ratio of fat-to-muscle weight of F1 males at 15 weeks of age. In (**f–i**), $n = 17$ in scrambled RNA group, $n = 10$ in synthetic tsRNAs group, $n = 17$ in sperm RNAs, and $n = 7$ in control. Statistical analysis was performed by two-tailed, two-way ANOVA (**b**, **f**) or one-way ANOVA (**c–e**, **g-i**), uncorrected Fisher's LSD. Data are show as mean ± SEM. Source data are provided as a Source Data file.

compared with those derived from natural mating. This alteration may attribute to the injection of random RNA into the male pronuclei of zygotes combined with embryo transfer.

Most evidence that tsRNAs-mediate the transmission of paternal traits to offspring is based on long-term environmental exposure spanning from spermatogenesis in the testis and sperm transition through the epididymis[1–3,19,20]. A recent report indicated that the composition of human sperm tsRNAs is acutely sensitive to dietary shifts[51]. Because sperm is stored in the epididymis prior to ejaculation, sperm tsRNAs may be vulnerable to environmental stimuli during the transition through the epididymis. In this study, inflammation was induced in the F0 mice for only seven days, a length of time that would just span sperm transit from the caput to the cauda epididymis[52]. Our results suggest that sperm tsRNAs are vulnerable to environmental stimuli when transitioning through the epididymis, and this exposure is sufficient to alter the phenotype of the offspring. The epididymis can be exposed to a constant risk of inflammatory conditions resulting from both infectious and noninfectious causes[15]. Interestingly, most of the environmental inputs that are reported to alter metabolic phenotype in offspring (physical trauma, unhealthy diet, toxin exposure, and lifestyle factors) also elevate inflammatory markers in the epididymis[8,15,53–56]. For example, high fat diet-induced gut microbiota can exacerbate chronic systematic inflammation and inflammation in the epididymis in human and rodent models[53,54]. Toxins such as nicotine, aflatoxin-B1, LPS, and ethanol can induce inflammation in the epididymis[55–58]. Therefore, further studies should seek to clarify whether inflammation in the epididymis functions as a nexus for the transmission of parental environmental stressors to offspring.

Taken together, our data identify a role for paternal inflammation in regulation of the sperm epigenome and metabolic phenotypes in offspring. We show that (i) paternal inflammation can influence offspring metabolic phenotype via information in sperm; and (ii) Ang is essential in the inflammation-induced reshaping of the sperm tsRNAs profile. Further exploration of these mechanisms will lead to a deeper understanding of the 'information capacity' mediated by versatile combinations of sperm tsRNAs.

## Methods

**Animals**. The C57BL/6 mice were maintained and bred in specific pathogen-free condition at Laboratory Animal Center of Nanjing Agricultural University and were allocated to experimental groups on the basis of their genotype and randomized into different treatment groups. Generation and characterization of $Ang^{-/-}$ mice were previously described[21]. All studies were performed according to experimental protocols approved by the Institutional Animal Care and Use Committee (IACUC) of Nanjing Agricultural University, and all procedures were conducted in accordance with the "Guidelines on Ethical Treatment of Experimental Animals" (2006) No. 398 set by the Ministry of Science and Technology,

China and the Regulation regarding the Management and Treatment of Experimental Animals" (2008) No. 45 set by the Jiangsu Provincial People's Government.

For inflammatory model, 8-week-old male mice were received intraperitoneal injections of LPS (10 mg/kg BW), once every other day, for a total of four injections. Control mice were injected with the same volume (100 μL) of saline. LPS derived from *E. coli* (strain O111:B4) were purchased from Sigma-Aldrich. All dilutions were conducted in 0.9% NaCl (w/v) in endotoxin free water. Each male was mated with normal females (aged 8 weeks) after last saline or LPS injection for 2 days. To confirm the inflammatory responses in mouse models, mice were killed 12 h after single LPS or saline treatment. The blood was taken from mouse orbit after anesthesia and centrifuged to separate serum. The interleukin 6 (IL6) and interleukin 10 (IL10) in the serum and supernatant of caput epididymis were quantitated using a double antibody enzyme-linked immunosorbent assay (ELISA) method. The mouse IL6 ELISA kit (ELM-IL6-1) and IL10 ELISA kit (ELM-IL10) were purchased from RayBiotech, Inc. (Norcross, GA, USA).

**Blood glucose examination during glucose tolerance test (GTT)**. During GTT, an intraperitoneal injection of glucose with a single dose of 2 g/kg body weight was performed in 12 h-fasted mice. Blood samples were collected from the tail vein before glucose injection (0 min) and at 15, 30, 60, and 120 min afterward. Blood glucose concentration was immediately measured by a glucose meter (ACCU-CHEK Active Blood Glucose Meter, Roche).

**Blood glucose examination during insulin tolerance test (ITT)**. ITT was performed by intraperitoneal injection of insulin (0.75 IU/kg, Aladdin, CAS 12584-58-6). Blood glucose concentrations were measured before insulin injection (0 min) and at 15, 30, 60, and 120 min after insulin injection. Blood samples were collected from mice tail vain and blood glucose concentration was immediately measured by a glucose meter.

**Blood glucose examination during pyruvate tolerance test (PTT)**. During PTT, an intraperitoneal injection of pyruvate with a single dose of 1.5 g/kg body weight was performed in 12 h-fasted mice. Blood samples were collected from the tail vein before pyruvate injection (0 min) and at 15, 30, 60, and 120 min afterward. Blood glucose concentration was immediately measured by a glucose meter.

**Histological analysis**. For histomorphological evaluation, a portion of fixed perigonadal fat tissue was dehydrated, embedded in paraffin and sectioned at 5 μm. The sections were stained with hematoxylin/eosin (H&E). The cross-sectional area of adipocyte cells were examined under a microscope (BX63F OLYMPUS Micro Image System, OLYMPUS, Tokyo, Japan).

**Running experiment**. Mice were acclimated to the treadmill as described previously[59]. Before treadmill tests, mice were exposed to the treadmill. Training started at 10 m/min for 10 min and 22 m/min for 15 min; the following day, the treadmill test was performed. Exercise capacity was estimated by determining the all-out running speed on the treadmill when mice could not or would not run forward despite any stimuli for 20 s. The protocol consisted of a starting speed of 10 m/min for 10 min, followed by an increase of 2 m/min every 30 s.

**Protein extraction and western blot analysis**. For total protein extraction, gastrocnemius muscle or caput epididymis were homogenized in RIPA buffer (50 mM Tris-HCl pH 7.4, 150 mM NaCl, 1% NP40, 0.25% Na-deoxycholate, 1 mM PMSF, 1 mM sodium orthovanadate with Roche EDTA-free complete mini protease inhibitor cocktail, no. 11836170001). For membrane protein extraction, gastrocnemius muscles were cut into thin strips and incubated (1 h, 34 °C, 100 rpm) in 140 mM KCl, 10 mM MOPS (pH 7.4) containing collagenase VII (150 units/mL) and aprotinin (10 mg/mL). The muscle was washed with KCl/MOPS containing 10 mM $Na_2EDTA$ (pH 7.4), and the supernatants were combined. Percoll (3.5%),

KCl (28 mM), and aprotinin (10 µg/mL) were added to the supernatant. This solution was layered under 3 mL of 4% Nycodenz and 1 mL of KCl/MOPS and centrifuged (60 × g, 45 min, 25 °C). The vesicles were harvested from the interface of Nycodenz and KCl/MOPS and pelleted by centrifugation (9,000 × g, 10 min, 25 °C), and the resulting pellet was resuspended in KCl/MOPS and stored at −80 °C for Western blotting. The protein concentration was measured with the BCA Protein Assay Kit (Pierce, Rockford, IL, USA) according to a previous publication. Forty micrograms of protein extract were used for electrophoresis on a 15% or 10% SDS-PAGE gel. The rabbit Anti-GLUT4 antibody (1:1,000, BS3680, Bioworld), rabbit Anti-Ang antibody (1:200, home-made) were used as primary antibody. Protein loading controls for each experiment used rabbit anti-GAPDH antibody (1:10,000, MB001H, Bioworld, China) and rabbit Anti-Na$^+$/K$^+$-ATPase antibody (1:500, BS90909, Bioworld, China). All the operations were carried out according to the recommended protocols provided by the manufacturers.

**Immunohistochemistry on cryosections for GLUT4 analysis.** Freshly muscles collected from mice were fixed with 4% paraformaldehyde for 2 h, then precipitated with 30% sucrose overnight. The fixed tissue was embedded in OCT and frozen sections were taken. Slices were treated with TBS containing 0.5% Triton X-100 at room temperature for 1 h, blocked with 10% FBS and then immunostained with a rabbit anti-mouse GLUT4 antibody (1:1,000, BS3680, Bioworld, China), overnight at 4 °C. The Cy3-labeled Goat Anti-Rabbit IgG (H + L) (1:500, A0516, Beyotime, China) was used as the second antibody. The sections were counterstained with DAPI to visualize cell nuclei. Negative control sections were incubated with 10% FBS instead of primary antibodies. Slides were visualized using a fluorescence microscope (Leica, DMI6000 B, Germany).

**RNA-seq.** Total RNA was extracted from mouse gastrocnemius muscle. Libraries of template molecules suitable for strand-specific high-throughput RNA sequencing were created using a TruePrepTM DNA Library Prep Kit V2 for Illumina® (Vazyme, Nanjing, China). The libraries were sequenced on Illumina NovaSeq 6,000 sequencer as paired-end 150 bp reads following Illumina's instructions. Image analysis and base calling were performed using RTA 2.7.3 and bcl2fastq 2.17.1.14. Adapter dimer reads were removed using DimerRemover v0.9.2. Reads were mapped to the mouse genome (mm9) using Tophat v2.0.14 and Bowtie v2-2.1.0. Quantification of gene expression was performed using HTSeq v0.6.1 and gene annotations from Ensembl release.

**RT-PCR and quantitative RT-PCR.** Total RNA was extracted from the gastrocnemius, testis, caput epididymis and cauda epididymis using TRIzol reagent (Invitrogen, cat. no. 15596026) according to the manufacturer's instructions. Total RNA was processed to remove genome DNA using RQ1 RNase-Free DNase (Promega, cat. no. M6101). Then, 1 µg of RNA was reverse transcribed using the M-MuLV Reverse Transcriptase Reaction system (NEB, cat. no. M0253L). cDNAs obtained were diluted and used for quantitative PCR (qPCR). Each qPCR assay was performed with a standard dilution curve of a calibrator, for a mixture of different cDNA, to precisely quantify relative transcript levels. Gene-specific primers were used with SYBR green (Promega, cat. no. A6002) for detection on a LightCycler 480 system (Roche). The primer sequences used were synthesized by Genewiz as shown in Supplementary Tab. 6.

**Sperm sample collection and RNA extraction.** Mature sperm were isolated from the cauda epididymis of Ang$^{+/+}$ Con, Ang$^{+/+}$ Inf, Ang$^{-/-}$ Con, and Ang$^{-/-}$ Inf mice and processed for RNA extraction as previously described[1]. In brief, sperm were released from cauda epididymis into 5 mL phosphate-buffered saline (PBS) maintained at 37ºC for 15 min incubation, after which were then filtered with 40 µm cell strainer to rid of the tissue debris. The sperm were then treated with somatic cell lysis buffer (0.1% SDS, 0.5% Triton X in DEPC H$_2$O) for 40 min on ice to eliminate somatic cell contamination, after which the sperm were pelleted by centrifugation at 600 × g for 5 min. After removal of suspension, the sperm pellet was resuspended and washed twice in 10 mL of PBS then pelleted at 600 g for 5 min. The sperm pellet were added with TRIzol reagent, homogenized, followed by RNA extraction.

**Deep sequencing and quality control.** Small RNA libraries were constructed according to MGIEasy Small RNA Library Prep Kit (MGI, Shenzhen, China), the small RNA libraries were prepared followed by library quality validation for sequencing. All RNA libraries preparation and quality examination were performed by BGI. For each RNA library, 10 million reads (raw data) were generated by BGISEQ-500. Sequence reads that fit any of the following parameters were removed with the following standard quality control criteria: (1) The reads with N; more than 4 bases whose quality score is lower than 10 or more than 6 bases whose quality score is lower than 13. (2) The reads with 5′ primer contaminants or without 3′ primer. (3) The reads without the insert tag. (4) The reads with ploy A. 5) The reads shorter than 18 nt. The clean reads was obtained after data filtration.

**Small RNA annotation.** Small RNA sequences were annotated using the pipeline SPORTS (Small non-coding RNA annotation Pipeline Optimized for rRNA- and tRNA-Derived Small RNAs, https://github.com/junchaoshi/sports1.0). Small RNA tags were annotated with miRNA, tRNA, rRNA and other small non-coding RNA from miRBase19, Genbank and Rfam databases using blastn with standard parameters: −F F −e 0.01. rRNA reads were removed for length distribution analysis. To analyze differential expression of small RNAs between Ang$^{+/+}$ Con, Ang$^{+/+}$ Inf, Ang$^{-/-}$ Con, and Ang$^{-/-}$ Inf mice sperm, tsRNA reads were normalized to RPM (reads per million) or RPM tsRNA reads respectively. The P value and q value between samples were generated by DEGseq package of R. Those small RNAs that had P value smaller than 0.05 and had the fold change number larger than 2 were labeled as significantly changed RNAs.

**Northern blots.** RNA was extracted from caudal sperm or caput epididymis and separated by 15% urea-PAGE. Gels were stained with SYBR Gold, and immediately imaged and transferred to nylon membranes (GE Amersham, cat. no. MRPN303B), and UV crosslinked with an energy of 0.12J. Membranes were pre-hybridized with DIG pre-hybridization (Ambion: ULTRAhyb® Ultrasensitive Hybridization Buffer REF: AM8670) for at least 1 h at 42 °C. The membranes were incubated overnight (12–16 h) at 42 °C with DIG-labeled oligonucleotides probes synthesized by GENEWIZ, Inc. as shown in Supplementary Tab. 7. Probes were added to the hybridization solution at a final concentration of 16 nM, and incubated overnight. The membranes were washed twice with low stringent buffer (2× SSC with 0.1% (wt/vol) SDS) at 42 °C for 15 min each, then rinsed twice with high stringent buffer (0.1× SSC with 0.1% (wt/vol) SDS) for 5 min each, and then finally rinsed in washing buffer (1× SSC) for 10 min. Following the washes, the membranes were transferred into 1× blocking buffer (Roche REF:11096176001) and incubated at room temperature for 2–3 h, after which the DIG antibody (Roach: Anti-Digoxigenin-AP Fab fragments, REF: 11093274910) was added into the blocking buffer at a ratio of 1:10,000 and incubated for an additional half hour at room temperature. The membranes were then washed four times in DIG washing buffer (1× maleic acid buffer, 0.3% Tween-20) for 15 min each, rinsed in DIG detection buffer (0.1 M Tris-HCl, 0.1 M NaCl, pH 9.5) for 5 min, and then coated with CSPD ready-to-use reagent (Roach REF: 11755633001). The membranes were incubated in the dark with the CSPD reagent for 15 min at 37 °C before imaging using a Bio-Rad imaging system.

**Isolation of 30–40 nt RNAs from sperm total RNAs.** Sperm total RNAs are extracted by Trizol (Invitrogen). 1–2 µg total sperm RNAs were separated by denatured 15% PAGE with 7 M urea. The gel was stained with SYBR Green II solution (Invitrogen). The location of RNA fractions was determined by the position of standard small RNA markers by using long-wave UV light illumination of the gel. Small RNAs sized at 30–40 nt were excised from the gel as previously described[1,2].

**Oocytes and zygotes collection.** Embryo collection and transfer were performed as previously described. In brief, virgin female mice aged 6 weeks were selected as oocyte donors for superovulation, which were performed by intraperitoneal injection with 7.5 IU pregnant mare serum gonadotropin (PMSG), after 48 h intraperitoneal injection with 5 IU human chorionic gonadotrophin (hCG). Oocytes were collected 12-18 h after hCG administration, and zygotes were collected from the successfully mated female mouse.

**Zygote 30–40 nt RNAs and synthetic RNAs microinjection and embryo transfer.** MII (first polar body present) oocytes were used to perform intracyto-plasmic sperm injection (ICSI) and the fertilization was confirmed by the presence of two pronuclei. Small RNAs isolated from LPS-treated or saline-treated mice sperm total RNAs with the nucleotide length ranging from 30–40 nt, chemically synthetic 5′ end phosphorylated tsRNAs sequences or synthetic scrambled RNA (Supplementary Tab. 4), with a concentration of 2 ng/µL, or RNase-free water as control were microinjected into fertilized eggs of C57BL/6 background) performed as previously described[1–3,9]. This amount equals approximately the total RNA of 10 sperm according to previous report[1,2]. Chemically synthetic tsRNAs and scrambled RNA were synthesized by Genewiz. All RNAs were injected into the male pronuclei of zygotes using a Leica microinjection system. The zygotes were then transferred to the oviduct of surrogate mother of C57BL/6 background.

**Statistics and reproducibility.** The offspring growth rate during the period 3–12 weeks in Fig. 1a was analyzed using a MIXED procedure for repeated measures (SAS 9.4), with paternal treatment, age, litter size and the interaction between paternal treatment and age entered as fixed factors, and father as a random factor, repeated age (subject = mother/type = un). GraphPad Prism 7 was used to analyze data for mouse body weight, GTT, ITT, PTT, qPCR and ELISA. Data are presented as mean ± SEM. By two-tailed unpaired Student's t-test for Fig. 1b, f, i, and 2a and Supplementary Fig. 1a–e, g. By two-way ANOVA with uncorrected Fisher's LSD for Figs. 1c–e, 2f, 4b, f and Supplementary Figs. 3d, 6a, 7a. By one-way ANOVA with uncorrected Fisher's LSD for Fig. 2d, e, g, h, 3k–n, 4c–e, g–i and Supplementary Figs. 2a–l, 3a, b, e, f–j, 6b–j, 7b–j. n represents the number of mice used in each group, depending on the availability of mice. The number of experiments is detailed in the figure legends. Four independent sets of RNA samples were used for transcriptome sequencing (Fig. 1l, Supplementary Fig. 1h, i). Three independent

sets of sperm RNA samples were used for small RNA sequencing (Fig. 3a–n, Supplementary Fig. 4). In Fig. 1g, h, j, k; 3o and Supplementary Fig. 3c, n = 5–6 mice were examined as indicated in the figure legends (5 slices per mouse for each staining). Figure 3o and Supplementary Fig. 5 are representative of three independent experiments with similar results. Figure 2b and Supplementary Fig. 1f, are representative of two independent experiments with similar results.

**Reporting summary**. Further information on research design is available in the Nature Research Reporting Summary linked to this article.

## Data availability

The small RNA-seq and transcriptome sequencing data generated in this study have been deposited in the NCBI Sequence Read Archive (SRA) database under the BioProject accession number PRJNA687877. The data supporting this study are available in the Article, Supplementary Information, or available from the corresponding authors upon reasonable requests. Source data are provided with this paper.

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

## Acknowledgements

The authors thank Dr. Qi Chen and Dr. Junchao Shi (University of California, Riverside) for their help on small RNA-seq analysis and critical suggestions and comments. This work was supported by the National Natural Science Foundation of China (31872436, 32072807), the Natural Science Foundation of Jiangsu Province (BK20181323) and the Priority Academic Program Development of Jiangsu Higher Education Institutions.

## Author contributions

B.H. and Y.Z. conceived the idea and designed experiments. B.H. and Y.Z. wrote the main manuscript and integrated inputs from all authors. Y.Z., L.R., X.S., Z.Z. and J.Y. performed the mouse breeding, embryo manipulation related experiments and phenotype analyses and under the supervision of B.H. and R.Z. Y.Z. prepared sperm RNA samples and performed bioinformatics analyses for small RNA-seq and transcriptome data with help from J.L. under the supervision of B.H. and Y.J. Y.Z., L.R., X.S., Z.Z. and Y.X. performed the histological analysis, Northern blot, Western blot and qRT-PCR. B.H. communicated with the editor and coordinated communications with R.Z., J.S. and G.H. who supervised different aspects of the paper.

## Competing interests

The authors declare no competing interest.
