## [Peer Review File · Nature Communications]

Reviewer comments, second round review -

Reviewer #1 (Remarks to the Author):

This manuscript by Zhang et al. describes a possible new function of 5'-tsRNAs in metabolic disorders of mouse offspring. The authors show that Ang gene is responsible for alteration of 5'-tsRNAs expression profile caused by inflammation. The authors concluded that 5'-tsRNAs contribute to parental induced metabolic disorders in offspring.

Critique.

In general, this manuscript lacks molecular details. No attempts were made to figure out what is happening on molecular level.

The injection of the fraction of RNA containing "mostly" 5'-tsRNAs is problematic since there may be many different types of small RNAs in the fraction, and it is unclear which RNAs caused the effect authors described.

Even if 5'-tsRNAs are responsible for the effect, this group of small RNAs is a very large group. Then the question is are all of these RNAs, or some of these RNAs, or may be only one of these RNAs cause the effect.

Did the authors try to synthesize 5, 10, or 20 of 5'-tsRNAs, mix them and see if these RNAs cause the effect they describe?

Reviewer #2 (Remarks to the Author):

It is of general interest to clarify if the "sperm RNA code" mediates paternal inflammation reprogramming of offspring phenotype, and identify the key genetic factor involving in this process. In this manuscript, Zhang et al. adopted a LPS-instigated inflammation mouse model to investigate the effect of paternal inflammation on the metabolic health of adult offspring, and found that paternal inflammation can influence offspring metabolic phenotype via information in sperm, and angiogenin (Ang), a known tsRNA generating enzyme, is essential in the inflammation-induced reshaping of the sperm tsRNAs profile. Especially, previous reports showing tsRNAs-mediated transmission of paternal traits to offspring have been mainly using long-term environmental exposure spanning from spermatogenesis in the testis and sperm transition through the epididymis. In this study, inflammation was induced in the F0 mice for only seven days, a length of time that just spans sperm transit from the caput to the cauda epididymis, demonstrating even a short paternal environmental insult is sufficient to alter the phenotype of the offspring. Therefore, this work would advance our understanding on transgenerational inheritance and on Ang and RNA biology. In general, this study is well-designed, and the methodology is sound. I would like to suggest:

1. Regarding Ang regulates tsRNAs during sperm development: (i) since Ang is not expressed at particularly high level in the epididymis, it needs more stronger evidence to support its role in sperm transition. Now the authors only presented the Ang mRNA levels in the testis, caput epididymis, and cauda epididymis. It would be necessary to detect the protein levels as well. Especially, could Ang be upregulated throughout the 7d induction? (ii) It would be interesting to compare the composition

difference of tsRNAs between the caput and the cauda of epididymis in Ang^{+/+} and Ang^{-/-} males. Currently the authors only analyzed the tsRNAs of cauda. (iii) Now that the authors considered that 5'-tsRNA-Gly-GCC, 5'-tsRNA-Val-TAC, 5'-tsRNA-iMet-CAT, 5'-mt-tsRNA-Val-TAC, and 5'-mt-tsRNA-Met-CAT were the key inflammation-induced tsRNAs, it would be necessary to investigate their contributions in passing paternal information to the offspring.

2. The authors need to show that the LPS treatment did induce inflammation in testis. It has been reported that Ang is involved in myeloid cell differentiation by one of the coauthors (Guofu Hu) of this manuscript. Is it possible that deletion of Ang affects the response of mouse to LPS treatment? i.e., making the Ang^{-/-} paternal mice less inflammatory?

Minor comments :

1. I do not understand "Strikingly, we found comparable glucose intolerance phenotype between Ang^{-/-} and Ang^{+/+} male offspring, with Ang^{+/+} Inf F1 males showing significantly higher glucose than the other F1 males as assayed by GTT, but not ITT (Fig. 2d-g)."

2. There are some misspelled words in the manuscript, such as "menberane" in Supplementary Fig. S1f.

Reviewer #3 (Remarks to the Author):

Zhang et al have present a studying assessing the effects of the tRNA RNase Ang on the transmission of metabolic traits to offspring from fathers with inflammation. They showed that male mice with inflammation who produced male offspring with altered glucose tolerance and increased adiposity had increase Ang expression in sperm isolated from the caput epididymis. The effect of founder inflammation on offspring metabolic outcomes was abolished in both Ang^{-/-} males with and offspring produced from microinjection of sperm fractions from Ang^{-/-} males. I would like to commend the authors for the amount of work presented in the publication which advances understanding of how paternal stress program offspring pheonotypes, however further clarification is required.

Major Comments:

- I found the methods lacked information about key components of the experimental plan, that without, make it hard to reviewer for validity. Examples;

o While the authors provide n-values of the animals used in the figure legends, they do not provide any information about the number of founders used to generate offspring, how many offspring were sampled per litter how many founders are represented in the offspring numbers.

o No information is provided about the ages of when the animals were mated or any information about the C57/B6J inflammatory model.

o Confirmation of number of mums that embryos were transferred into for the microinjection experiments – was it 10 per group or 5 per group (i.e. 10-15 per oviduct – was only one oviduct transferred too, or did both oviducts get transfers). You also only put in live birth rates and not how many mothers were pregnancy or average litter number of the mothers (sup table 1). If all your offspring came from one or two mothers than how did you control for the mother effect. Etc.

- Why didn't the authors look at the tsRNA expression profiles of cauda sperm in their c57/B6J inflammatory model similar to what they did in their Ang^{-/-} model and compared the differences/similarities especially if they are saying that this is the causal pathway in which inflammation is modifying offspring metabolic phenotype through increased Ang expression in the epididymis?

- For their microinjection experiments, why did the authors choose to inject the amount of tRNA that is equivalent to 10 sperm (2ng/μl) when only one sperm contributes ~10-30fg total of RNA of which tRNAs is a subset and the fact that they had previously been fertilised by sperm (who have already contributed a tRNA pool). This doesn't seem biological. Can authors explain why they didn't use a scrambled RNA control, especially when live rates were so much higher in the sham injection (70%, vs 20-40%, sup table 1) in your injection group, suggesting that injection of that amount of RNA is detrimental to embryo development, so I do not agree with their statement on lines 174-177.
- The most interesting comparison of tRNAs is between the Ang +/+ inflammation and Ang -/- inflammation group (Sup table 2), yet no comparison between these two. Also not all tRNAs were restored by Ang -/- (sup table 2) which suggests that there are other RNase involved in producing tRNA halves and potentially in the inflammatory paternal mediated offspring phenotype. This was actually found in publication from Su et al (reference 29) 'Angiogenin generates specific stress-induced tRNA halves and is not involved in tRF-3-mediated gene silencing', which found that the majority of stress-induced tRNA halves, except for the 5' half from tRNA^{His}GTG and the 3' half from tRNA^{Asp}GTC, are ANG independent, suggesting there are other RNases that produce tRNA halves. They also found that a group of tRNAs (tRNA^{Gly}, tRNA^{Glu}, tRNA^{Lys}, tRNA^{Val}, tRNA^{His}, tRNA^{Asp}, and tRNA^{Sec}) were more responsive to Ang. The authors may want to think about modifying the graphs in Figure 3 to be more representative of those tRNAs that are more responsive to Ang. But also indicate in text that there are likely other mediators.
- Not all offspring measure are completed in each of the models. It make it seem unfinished. For instance;
 - o RNA transcription of muscular tissue is presented in the very first model (C57/Bl6 inflammatory model) but not in the other two models, even if they measured the top changing genes.
 - o Fat phenotypes, running speed etc. not present in the microinjection models.
- Statistical analysis = The statistical analysis should contain mixed models to include both the mother and father in which the offspring were generated from (if more than on offspring was sampled per litter – can't tell) as well as litter size if litter sizes were not normalised. This will ensure that what you are seeing is not biased because of one litter etc.

Minor Comments:

- Add concentration of HCG use for superovulation to methods.
- Even though you reference a previous paper, you may want to put a table of the founder phenotypes, mating rates, litter sizes from your C57BL/6J inflammatory model and a separate one for your Ang -/- models.
- The discussion is a little light on, more information on how Ang is modifying the tRNA pool of sperm through the epididymis transit I think is important to speculate how modifications to tRNA sperm pool are causing their effect at fertilisation and therefore offspring phenotypes could be expanded.
- It would benefit the reader if you put the group labels under each figure.
- If required the muscle transcriptional profiles could go as supplementary.

Response to the Reviewers ' Comments

We thank the reviewers for the thoughtful and positive review of the manuscript. We conducted additional experiments and revised the manuscript carefully to address the questions and comments raised by the reviewers. Please refer to the reply to specific comments for details. We hope that the reviewers will find our revised manuscript acceptable for publication.

Reviewer #1 (Remarks to the Author):

This manuscript by Zhang et al. describes a possible new function of 5'-tsRNAs in metabolic disorders of mouse offspring. The authors show that Ang gene is responsible for alteration of 5'-tsRNAs expression profile caused by inflammation. The authors concluded that 5'-tsRNAs contribute to parental induced metabolic disorders in offspring.

Critique.

In general, this manuscript lacks molecular details. No attempts were made to figure out what is happening on molecular level.

Reply: Thank you for the questions relating to some important points that we glossed over in the original submission. We conducted additional experiments and revised the manuscript carefully to address the questions and comments raised by the reviewer.

The injection of the fraction of RNA containing “mostly” 5'-tsRNAs is problematic since there may be many different types of small RNAs in the fraction, and it is unclear which RNAs caused the effect authors described.

Reply: Thanks for the suggestions. Recent, studies have found that sperm 30-40 nt RNA fractions and its modifications can be rapidly altered by environmental inputs and contribute to intergenerational inheritance of acquired metabolic disorders (Chen et al., 2016; Sarker et al., 2019; Zhang et al., 2018). As indicated by the reviewer, there are many different types of sncRNAs in sperm 30-40 nt RNA fractions (tsRNAs, rsRNAs, and so on) (Chen et al., 2016; Chu et al., 2017; Sharma et al., 2016; 2018). The 5'-tsRNAs (30-35 nt) is extremely enriched in mature sperm (Chen et al., 2016; Peng et al., 2012; Sharma et al., 2016), but the biogenesis and function of these RNAs remain unclear.

Angiogenin (Ang) is a stress-responsive RNase (also known as RNase 5) which could cleave tRNAs at the anticodon loop for fragmentation into 5'-tsRNAs (30-35 nt) and 3'-tsRNAs (40-50 nt). In the present study, we used an *Ang*-deletion mouse model and microinjection method to demonstrate whether Ang-mediated biogenesis of tsRNAs in sperm contributes to paternal inflammation-induced metabolic disorders in offspring. Our

previous results indicated that *Ang* deletion abolished paternal inflammation-induced metabolic disorders in offspring and prevented the inflammation-induced alteration of 5'-tsRNAs expression profile in sperm. Zygotic injection of sperm 30-40 nt RNA fractions (predominantly 5'-tsRNAs) from *Ang*^{+/+} Inf but not *Ang*^{-/-} Inf mice induced metabolic changes in the resultant offspring. These converging clues suggest that Ang-mediated alteration of sperm 30-40 nt RNA profile contributes to paternal inflammation-induced metabolic disorders in offspring.

To further determine whether 5'-tsRNAs could mimic the function of endogenous sperm 30-40 nt RNA fractions, we synthesized a combination of the most highly expressed 5'-tsRNAs whose abundances displayed increases in inflammatory mouse sperm and decreased by *Ang* deletion to baseline levels in the sperm (Figure 3j, Supplementary Figure 4 and Supplementary Table 5) for zygotic injection. The F1 male offspring from the sperm 30-40 nt RNA fractions (sperm RNAs) group showed impaired glucose tolerance, as evidenced by significantly higher glucose levels assayed by GTT compared with the levels in scrambled RNA and synthetic tsRNAs group, and the control group (Figure 4f-g and Supplementary Figure 6k-l). Strikingly, the F1 males of synthetic tsRNAs group showed a slight, but significant increase of glucose levels compared to those of the scrambled RNA group (Figure 4f-g). Moreover, the resultant offspring from the synthetic tsRNAs injection group displayed higher bodyweight compared with the levels in scrambled RNA, sperm RNAs, and control groups (Figure 4h). Consistent with these findings, mice in the synthetic tsRNAs and sperm RNAs group displayed an elevation of perigonadal fat mass (Supplementary Figure 6m) and fat-to-muscle ratio (Figure 4i), accompanied by an increase in the mean cross-sectional area of adipocyte cells (Supplementary Figure 6n). There was no significant difference in gastrocnemius muscle mass (Supplementary Figure 6o), but the synthetic tsRNAs injection group exhibited lower treadmill running capacity compared with the levels in scrambled RNA, sperm RNAs groups, as well as the control group (Supplementary Figure 6p). Together, the results suggested that zygotic injection of a mixture of synthetic 5'-tsRNAs can partly induce metabolic phenotypes in offspring.

In conclusion, our findings demonstrate that Ang-mediated biogenesis of 5'-tsRNAs in sperm contributes to paternal inflammation-induced metabolic disorders in offspring.

References:

- Chen, Q. *et al.* Sperm tsRNAs contribute to intergenerational inheritance of an acquired metabolic disorder. *Science* **351**, 397-400 (2016).
- Chu, C. *et al.* A sequence of 28S rRNA-derived small RNAs is enriched in mature sperm and various somatic tissues and possibly associates with inflammation. *Journal of Molecular Cell Biology*, **9**, 256-259 (2017).
- Peng, H. *et al.* A novel class of tRNA-derived small RNAs extremely enriched in mature mouse sperm. *Cell research* **22**, 1609-1612 (2012).
- Sarker, G. *et al.* Maternal overnutrition programs hedonic and metabolic phenotypes across generations through sperm tsRNAs. *Proceedings of the National Academy of Sciences of the United States of America* **116**, 10547-10556 (2019).

- Sharma, U. *et al.* Biogenesis and function of tRNA fragments during sperm maturation and fertilization in mammals. *Science* **351**, 391-396 (2016).
- Sharma, U. *et al.* Small RNAs are trafficked from the epididymis to developing mammalian sperm. *Developmental Cell* **46**, 481-494 e486 (2018).
- Zhang, Y. *et al.* Dnmt2 mediates intergenerational transmission of paternally acquired metabolic disorders through sperm small non-coding RNAs. *Nature cell biology* **20**, 535-540 (2018).

Even if 5'-tsRNAs are responsible for the effect, this group of small RNAs is a very large group. Then the question is are all of these RNAs, or some of these RNAs, or may be only one of these RNAs cause the effect.

Reply: We gratefully thank the reviewer for her/his crucial point. Our sperm small RNA seq analysis showing that expression levels of 88 5'-tsRNAs were increased and those of 17 genes were decreased in *Ang*^{+/+} Inf compared with *Ang*^{+/+} Con mice (Figure 3i and Supplementary Table 3). Strikingly, several 5'-tsRNAs were upregulated in inflammatory mice but expression levels were not reversed by *Ang* deletion (Figure 3n, Supplementary Figure 4 and Supplementary Table 3). Then, we synthesized a combination of ten 5'-tsRNAs (including 5'-tsRNA-Gly, 5'-tsRNA-Glu, 5'-tsRNA-Val and 5'-tsRNA-iMet) whose abundances displayed increases in inflammatory mouse sperm and decreased by *Ang* deletion to baseline levels in the sperm for zygotic injection (Figure 3j, Supplementary Figure 4 and Supplementary Table 5). The results suggested that zygotic injection of a mixture of synthetic 5'-tsRNAs can partly mimic the function of sperm 30-40 nt RNA fractions from *Ang*^{+/+} Inf mice to induce metabolic phenotypes in offspring. It indicates that *Ang*-dependent biogenesis of 5'-tsRNAs in sperm may contribute to paternal inflammation-induced metabolic disorders in offspring.

Previous researches indicated that transfection of a pool of tsRNAs but not single tsRNA, downregulated the mitochondria oxidative phosphorylation and translation/ribosome pathways and promoted lineage differentiation in embryoid bodies and embryonic stem cells in mice (Krishna *et al.*, 2019; Shi *et al.*, 2021). The induction of a biased metabolic state in the early embryo might alter metabolites or mitochondrial function to further trigger a chain reaction that continuously influences the metabolic state (Zhang *et al.*, 2019). Thus, the phenotypic outcome of injecting 30-40 nt RNA fractions or a pool of synthetic tsRNAs in our studies could be a combinatorial effect and may relate to the function of tsRNAs in cell fate regulation in the early embryo. Further work is needed to explore how *Ang*-mediated biogenesis of 5'-tsRNAs in sperm is decoded in early embryos to control offspring phenotypes.

References:

- Krishna, S. *et al.* Dynamic expression of tRNA-derived small RNAs define cellular states. *EMBO Reports* **20**, e47789 (2019).
- Shi, J. *et al.* PANDORA-seq expands the repertoire of regulatory small RNAs by overcoming RNA modifications. *Nature cell biology* **23**, 424-436 (2021).

Zhang, Y., Shi, J., Rassoulzadegan, M., Tuorto, F. & Chen, Q. Sperm RNA code programmes the metabolic health of offspring. *Nature Reviews Endocrinology* **15**, 489-498 (2019).

Did the authors try to synthesize 5, 10, or 20 of 5'-tsRNAs, mix them and see if these RNAs cause the effect they describe?

Reply: We appreciate this constructive and insightful comment. According to reviewer's instructions, we synthesized a combination of ten tsRNAs whose abundances displayed increases in inflammatory mouse sperm and decreased by *Ang* deletion to baseline levels in the sperm (Figure 3j, Supplementary Figure 4 and Supplementary Table 5) for zygotic injection to investigate whether they could mimic the function of endogenous sperm tsRNAs performed as previously described (Chen et al., 2016; Zhang et al., 2018; Sarker et al., 2019; Wang et al., 2021). Moreover, with scrambled RNA and sperm 30-40 nt RNAs from *Ang*^{+/+} Inf mice (sperm RNAs) injection as control. The resultant offspring were assessed for body weight, fat mass, muscle mass and metabolic parameters (Supplementary Table 6).

The results are added in the revised manuscript. (Figure 4 and Supplementary Figure 6, Line 221-244)

“We next synthesized a combination of the most highly expressed 5'-tsRNAs whose abundances displayed increases in inflammatory mouse sperm and decreased by *Ang* deletion to baseline levels in the sperm (Figure 3j, Supplementary Figure 4 and Supplementary Table 5) for zygotic injection to investigate whether they could mimic the function of endogenous sperm tsRNAs. Scrambled RNA and sperm 30-40 nt RNAs from *Ang*^{+/+} Inf mice as generated above (sperm RNAs) and normal male mice were used as a control (Supplementary Table 6). The F1 male offspring from the sperm RNAs group showed impaired glucose tolerance, as evidenced by significantly higher glucose levels assayed by GTT compared with the levels in scrambled RNA and synthetic tsRNAs group, as well as the control group (Figure 4f-g and Supplementary Figure 6k-l). Strikingly, the F1 males of synthetic tsRNAs group showed a slight, but significant increase of glucose levels compared to those of the scrambled RNA group (Figure 4f-g). Moreover, the resultant offspring from the synthetic tsRNAs injection group displayed higher bodyweight compared with the levels in scrambled RNA, sperm RNAs, and control groups (Figure 4h). Consistent with these findings, mice in the synthetic tsRNAs and sperm RNAs group displayed an elevation of perigonadal fat mass (Supplementary Figure 6m) and fat-to-muscle ratio (Figure 4i), accompanied by an increase in the mean cross-sectional area of adipocyte cells (Supplementary Figure 6n). There was no significant difference in gastrocnemius muscle mass (Supplementary Figure 6o), but the synthetic tsRNAs injection group exhibited lower treadmill running capacity compared with the levels in scrambled RNA, sperm RNAs groups, as well as the control group (Supplementary Figure 6p). The expression level of *SOCS3* were higher in the sperm RNAs group than the scrambled RNA and the control group while the expression level of

SOCS2 were higher in the synthetic tsRNAs group than the control group (Supplementary Figure 6q-t). Together, the results show that zygotic injection of a pull of synthetic tsRNAs can partly induce metabolic phenotypes in offspring.”

It must be pointed out that the live born (% transfer) was significantly smaller in the synthetic tsRNAs than in the scrambled RNA injection group (17% vs 47%, $\chi^2 = 10.793$, $df = 1$, $P < 0.01$, Supplementary Table 6 and 9). Recent researches have clarified that transfection of synthetic tsRNAs promoted the differentiation of embryonic stem cells (Krishna et al., 2019) and lineage in embryoid bodies (Shi et al., 2021). Fewer live fetuses per litter in synthetic tsRNAs group was probably due to synthetic tsRNAs-induced alteration in embryonic development and embryonic mortality. To eliminate the effects of litter size on the growth and metabolic parameters, we adjusted the litters to 4 to 5 pups on postnatal day 2 (because there were 4 to 5 pups per litter in synthetic tsRNAs injection group). There was no significant difference in the body weight of males among groups at the time of weaning. By far, we cannot eliminate the effects of lower litter size on offspring's body weight in synthetic tsRNAs group.

References:

- Chen, Q. *et al.* Sperm tsRNAs contribute to intergenerational inheritance of an acquired metabolic disorder. *Science* **351**, 397-400 (2016).
- Krishna, S. *et al.* Dynamic expression of tRNA-derived small RNAs define cellular states. *EMBO Reports* **20**, e47789 (2019).
- Sarker, G. *et al.* Maternal overnutrition programs hedonic and metabolic phenotypes across generations through sperm tsRNAs. *Proceedings of the National Academy of Sciences of the United States of America* **116**, 10547-10556 (2019).
- Shi, J. *et al.* PANDORA-seq expands the repertoire of regulatory small RNAs by overcoming RNA modifications. *Nature cell biology* **23**, 424-436 (2021).
- Wang, Y. *et al.* Sperm microRNAs confer depression susceptibility to offspring. *Science advances* **7**, eabd7605 (2021).
- Zhang, Y. *et al.* Dnmt2 mediates intergenerational transmission of paternally acquired metabolic disorders through sperm small non-coding RNAs. *Nature cell biology* **20**, 535-540 (2018).

Reviewer #2 (Remarks to the Author):

It is of general interest to clarify if the “sperm RNA code” mediates paternal inflammation reprogramming of offspring phenotype, and identify the key genetic factor involving in this process. In this manuscript, Zhang et al. adopted a LPS-instigated inflammation mouse model to investigate the effect of paternal inflammation on the metabolic health of adult offspring, and found that paternal inflammation can influence offspring metabolic phenotype via information in sperm, and angiogenin (Ang), a known tsRNA generating enzyme, is essential in the inflammation-induced reshaping of the sperm tsRNAs profile. Especially, previous reports showing tsRNAs-mediated transmission of paternal traits to offspring have been mainly using long-term environmental exposure spanning from spermatogenesis in the testis

and sperm transition through the epididymis. In this study, inflammation was induced in the F0 mice for only seven days, a length of time that just spans sperm transit from the caput to the cauda epididymis, demonstrating even a short paternal environmental insult is sufficient to alter the phenotype of the offspring. Therefore, this work would advance our understanding on transgenerational inheritance and on Ang and RNA biology.

In general, this study is well-designed, and the methodology is sound. I would like to suggest:

Reply: We thank the reviewer for her/his overall positive comments in our work.

1.Regarding Ang regulates tsRNAs during sperm development:

(i) Since Ang is not expressed at particularly high level in the epididymis, it needs more stronger evidence to support its role in sperm transition. Now the authors only presented the Ang mRNA levels in the testis, caput epididymis, and cauda epididymis. It would be necessary to detect the protein levels as well. Especially, could Ang be upregulated throughout the 7 d induction?

Reply: As indicated by the reviewer, there was an upregulation in the expression of *Ang* in the caput epididymis but not testis or cauda epididymis. We took the advice and detected the protein level of Ang in the caput epididymis after intraperitoneal injections of LPS (10 mg/kg BW) for 24 h and 7 d (once every other day, for a total of four injections). The results showed that the levels of Ang protein were upregulated in the caput epididymis after LPS treatment.

The results are added in the revised manuscript as Figure 2b.

(ii) It would be interesting to compare the composition difference of tsRNAs between the caput and the cauda of epididymis in *Ang*^{+/+} and *Ang*^{-/-} males. Currently the authors only analyzed the tsRNAs of cauda.

Reply: Thanks for the suggestions. Recent researches have clarified that tsRNAs were first synthesized in the caput epididymis and then trafficked to maturing sperm via epididymosomes (Sharma et al., 2016; 2018). To determine whether the changes of tsRNAs in cauda sperm are similar to those in the caput epididymis in inflammatory and *Ang*^{-/-} model, we detected the abundances of the most highly expressed tsRNAs in cauda sperm including 5'-tsRNA-Gly, 5'-tsRNA-Glu, 5'-tsRNA-Val and 5'-tsRNA-Cys in the caput epididymis by Northern blot. The results showed that the abundances of 5'-tsRNA-Gly, 5'-tsRNA-Glu and 5'-tsRNA-Val were increased in the inflammatory caput epididymis and decreased by *Ang* deletion (Figure 3o and Supplementary Figure 5). This was consistent with sperm small RNA seq analysis showing higher levels of 5'-tsRNA-Gly, 5'-tsRNA-Glu and 5'-tsRNA-Val in cauda sperm of *Ang*^{+/+} Inf mice compared with *Ang*^{+/+} Con, *Ang*^{-/-} Inf and *Ang*^{-/-} Con males. Differently, the abundance of 5'-tsRNA-Cys was upregulated in inflammatory caput epididymis but was not reversed

by *Ang* deletion (Supplementary Figure 5). Interestingly, these results were consistent with the previous finding that the biogenesis of 5'-tsRNA-Gly, 5'-tsRNA-Glu and 5'-tsRNA-Val are *Ang*-dependent but that of 5'-tsRNA-Cys is *Ang*-independent (Su et al. 2019).

The results are added in the revised manuscript as Figure 3o and Supplementary Figure 5. (Line 179-193)

References

- Sharma, U. *et al.* Biogenesis and function of tRNA fragments during sperm maturation and fertilization in mammals. *Science* **351**, 391-396 (2016).
- Sharma, U. *et al.* Small RNAs Are Trafficked from the Epididymis to Developing Mammalian Sperm. *Developmental cell* **46**, 481-494 e486 (2018).
- Su, Z., Kuscu, C., Malik, A., Shibata, E. & Dutta, A. Angiogenin generates specific stress-induced tRNA halves and is not involved in tRF-3-mediated gene silencing. *The Journal of biological chemistry* **294**, 16930-16941 (2019).

(iii) Now that the authors considered that 5'-tsRNA-Gly-GCC, 5'-tsRNA-Val-TAC, 5'-tsRNA-iMet-CAT, 5'-mt-tsRNA-Val-TAC, and 5'-mt-tsRNA-Met-CAT were the key inflammation-induced tsRNAs, it would be necessary to investigate their contributions in passing paternal information to the offspring.

Reply: According to reviewer's instructions, we synthesized a combination of the most highly expressed 5'-tsRNAs whose abundances displayed increases in inflammatory mouse sperm and decreased by *Ang* deletion to baseline levels in the sperm (Figure 3j, Supplementary Figure 4 and Supplementary Table 5) for zygotic injection to investigate whether they could mimic the function of endogenous sperm tsRNAs performed as previously described (Chen et al., 2016; Zhang et al., 2018; Sarker et al., 2019; Wang et al., 2021). Moreover, with scrambled RNA and sperm 30-40 nt RNAs from *Ang*^{+/+} Inf mice (sperm RNAs) injection as control. The resultant offspring were assessed for body weight, fat mass, muscle mass and metabolic parameters (Supplementary Table 6).

The F1 male offspring from the sperm RNAs group showed impaired glucose tolerance, as evidenced by significantly higher glucose levels assayed by GTT compared with the levels in scrambled RNA and synthetic tsRNAs group, and the control group (Figure 4f-g and Supplementary Figure 6k-l). Strikingly, the F1 males of synthetic tsRNAs group showed a slight, but significant increase of glucose levels compared to those of the scrambled RNA group (Figure 4f-g). Moreover, the resultant offspring from the synthetic tsRNAs injection group displayed higher bodyweight compared with the levels in scrambled RNA, sperm RNAs, and control groups (Figure 4h). Consistent with these findings, mice in the synthetic tsRNAs and sperm RNAs group displayed an elevation of perigonadal fat mass (Supplementary Figure 6m) and fat-to-muscle ratio (Figure 4i), accompanied by an increase in the mean cross-sectional area of adipocyte cells (Supplementary Figure 6n). There was no significant difference in gastrocnemius muscle

mass (Supplementary Figure 6o), but the synthetic tsRNAs injection group exhibited lower treadmill running capacity compared with the levels in scrambled RNA, sperm RNAs groups, as well as the control group (Supplementary Figure 6p).

In conclusion, the results suggested that zygotic injection of a mixture of synthetic 5'-tsRNAs can partly induce metabolic phenotypes in offspring.

The results are added in the revised manuscript. (Line 221-244)

References:

- Chen, Q. *et al.* Sperm tsRNAs contribute to intergenerational inheritance of an acquired metabolic disorder. *Science* **351**, 397-400 (2016).
- Sarker, G. *et al.* Maternal overnutrition programs hedonic and metabolic phenotypes across generations through sperm tsRNAs. *Proceedings of the National Academy of Sciences of the United States of America* **116**, 10547-10556 (2019).
- Wang, Y. *et al.* Sperm microRNAs confer depression susceptibility to offspring. *Science advances* **7**, eabd7605 (2021).
- Zhang, Y. *et al.* Dnmt2 mediates intergenerational transmission of paternally acquired metabolic disorders through sperm small non-coding RNAs. *Nature cell biology* **20**, 535-540 (2018).

2. The authors need to show that the LPS treatment did induce inflammation in testis. It has been reported that Ang is involved in myeloid cell differentiation by one of the coauthors (Guofu Hu) of this manuscript. Is it possible that deletion of Ang affects the response of mouse to LPS treatment? i.e., making the *Ang*^{-/-} paternal mice less inflammatory?

Reply: As indicated by the reviewer, there was an upregulation in the expression of *Ang* in the caput epididymis but not testis or cauda epididymis. Following the reviewer's suggestion, we performed additional experiment to confirm whether *Ang* deletion could protect mice from LPS-induced inflammation, male mice were killed 12 h or 24 h after LPS treatment. The blood was taken from mouse orbit after anaesthesia and centrifuged to separate serum. The levels of pro-inflammatory cytokines (TNF α , IL6 and IL1 β), and anti-inflammatory cytokine (IL10) in serum, caput epididymis and testis were measured by ELISA and RT-PCR.

There was no significant difference in the levels of proinflammatory cytokines (IL1 β , TNF α and IL6) and anti-inflammatory cytokine (IL10) in serum and caput epididymis between LPS treated *Ang*^{+/+} and *Ang*^{-/-} mice. Thus, the results indicated that *Ang* deletion did not protect mice from LPS-induced inflammation in caput epididymis (Supplementary Figure 2a-1).

The results are added in the revised manuscript as Supplementary Figure 2. (Line 129-132)

Minor comments:

1. I do not understand “Strikingly, we found comparable glucose intolerance phenotype between $Ang^{-/-}$ and $Ang^{+/+}$ male offspring, with $Ang^{+/+}$ Inf F1 males showing significantly higher glucose than the other F1 males as assayed by GTT, but not ITT (Figure 2d-g).”

Reply: We apologize that the sentence was misleading and it has been rewritten in the revised manuscript. “Furthermore, $Ang^{+/+}$ Inf F1 males showed significantly higher glucose than the other F1 males as assayed by GTT (Figure 2f-g), but not ITT (Supplementary Figure 3d-e). However, $Ang^{-/-}$ Inf F1 males exhibited no significant difference in glucose tolerance, similar to what was observed for $Ang^{+/+}$ Con F1 and $Ang^{-/-}$ Con F1 mice (Figure 2f-g).” (Line 140-144)

2. There are some misspelled words in the manuscript, such as “menberane” in Supplementary Figure S1f.

Reply: Thanks for the suggestions, we revised typing errors carefully.

In the original submission, data in Figure 1b, i, Figure 2c, h, and Supplementary Figure 1d were displayed as mean \pm S.D. In the revised manuscript, all data are plotted as mean \pm s.e.m. For adipocyte area analysis, three slices per animal (n= 5 mice per group) was observed. So, we revised “n = 15” into “n = 5” in Supplementary Figure 1d. Each dot represents one mouse. Figures were changed into Figure 1b (Figure 1b), Figure 1c (Figure 1c), Figure 1f (Figure 1f), Figure 1i (Figure 1i), Figure 2d (Figure 2c), Figure 2e (Figure 2i), Figure 2f (Figure 2d), Figure 2g (Figure 2f), Figure 2h (Figure 2h), Figure 4b (Figure 4b), Figure 4c (Figure 4d), Figure 4d (Figure 4f), Figure 4e (Figure 4g), Supplementary Figure 1d (Supplementary Figure 1d), Supplementary Figure 3a (Supplementary Figure 2a), and Supplementary Figure 3b (Supplementary Figure 2c) in the revised manuscript (Numbers in brackets are refer to the previous version of the manuscript). With no change in the original data and scatter plot.

These changes do not affect any conclusions. All statistic source data are provided in Supplementary Table 9.

Reviewer #3 (Remarks to the Author):

Zhang et al have present a studying assessing the effects of the tRNA RNase Ang on the transmission of metabolic traits to offspring from fathers with inflammation. They showed that male mice with inflammation who produced male offspring with altered glucose tolerance and increased adiposity had increase Ang expression in sperm isolated from the caput epididymis. The effect of founder inflammation on offspring metabolic outcomes was abolished in both $Ang^{-/-}$ males with and offspring produced from microinjection of sperm factions from $Ang^{-/-}$ males. I would like to commend the authors for the amount of work presented in the publication which advances understanding of how paternal stress program offspring pheonotypes, however further clarification is required.

Reply: We thank the reviewer for the appreciation of our study and for the constructive criticisms.

Major Comments:

- I found the methods lacked information about key components of the experimental plan, that without, make it hard to reviewer for validity. Examples;

o While the authors provide n-values of the animals used in the figure legends, they do not provide any information about the number of founders used to generate offspring, how many offspring were sampled per litter how many founders are represented in the offspring numbers.

Reply: According to reviewer's instructions, the information has been incorporated into the supplementary materials as Supplementary Table 1, 2, 4, 6 and 9. One or two male offspring of the average body weight were selected from each litter for GTT, ITT and PTT. Two or three male offspring of the average body weight were selected from each litter for treadmill test, body weight and organ weight measurements.

o No information is provided about the ages of when the animals were mated or any information about the C57/B6J inflammatory model.

Reply: For inflammatory model, 8-week-old mice received intraperitoneal injections of LPS (10 mg/kg BW), once every other day, for a total of four injections. Control mice were injected with the same volume (100 μ L) of saline.

The information has been incorporated into the Materials and methods section. (Line 328-330)

The inflammatory mouse model was verified by elevation of inflammatory biomarkers, including TNF α , IL6, IL1 β and IL10 in both serum, caput epididymis and testis. No significant difference in the production of proinflammatory cytokines (IL1 β , TNF α and IL6) and anti-inflammatory cytokine (IL10) in serum and caput epididymis between LPS treated *Ang*^{+/+} and *Ang*^{-/-} mice indicated that *Ang* deletion did not protect mice from LPS-induced inflammation in caput epididymis (Supplementary Figure 2a-1).

The results were added in the revised manuscript in as Supplementary Figure 2.

o Confirmation of number of mums that embryos were transferred into for the microinjection experiments – was it 10 per group or 5 per group (i.e. 10-15 per oviduct – was only one oviduct transferred too or did both oviducts get transfers). You also only put in live birth rates and not how many mothers were pregnancy or average litter number of the mothers (sup table 1). If all your offspring came from one or two mothers than how did you control for the mother effect. Etc.

Reply: For embryo transfer, zygotes (10-15) were transferred into one side of oviduct, both oviducts get transfers, with an amount of 25 zygotes transferred each surrogate mother. Four mice in each group were used as surrogate mother.

We apologize for the mistakes in the previous MS that “numbers of injected zygotes (n=50), transferred zygotes (n=50) and live born (% transfer) (35 (70%)) in Control injection group” was wrong. In fact, the number of injected zygotes is 100, the number of transferred zygotes is 100 and live born (% transfer) is 35% in Control injection group.

Detailed data were provided in Supplementary Table 4, 6 and 9.

- Why didn't the authors look at the tsRNA expression profiles of cauda sperm in their c57/B6J inflammatory model similar to what they did in their *Ang*^{-/-} model and compared the differences/similarities especially if they are saying that this is the causal pathway in which inflammation is modifying offspring metabolic phenotype through increased Ang expression in the epididymis?

Reply: Thanks for the suggestions. Recent researches have clarified that tsRNAs were first synthesized in the caput epididymis and then trafficked to maturing sperm via epididymosomes (Sharma et al., 2016; 2018). To determine whether the changes of tsRNAs in cauda sperm are similar to those in the caput epididymis in inflammatory and *Ang*^{-/-} model, we detected the abundances of the most highly expressed tsRNAs in cauda sperm including 5'-tsRNA-Gly, 5'-tsRNA-Glu, 5'-tsRNA-Val and 5'-tsRNA-Cys in the caput epididymis by Northern blot. The results showed that the abundances of 5'-tsRNA-Gly, 5'-tsRNA-Glu and 5'-tsRNA-Val were increased in the inflammatory caput epididymis and decreased by *Ang* deletion (Figure 3o and Supplementary Figure 5). This was consistent with sperm small RNA seq analysis showing higher levels of 5'-tsRNA-Gly, 5'-tsRNA-Glu and 5'-tsRNA-Val in cauda sperm of *Ang*^{+/+} Inf mice compared with *Ang*^{+/+} Con, *Ang*^{-/-} Inf and *Ang*^{-/-} Con males. Differently, the abundance of 5'-tsRNA-Cys was upregulated in inflammatory caput epididymis but was not reversed by *Ang* deletion (Supplementary Figure 5). Interestingly, these results were consistent with the previous finding that the biogenesis of 5'-tsRNA-Gly, 5'-tsRNA-Glu and 5'-tsRNA-Val are Ang-dependent but that of 5'-tsRNA-Cys is Ang-independent (Su et al. 2019).

The results are added in the revised manuscript as Figure 3o and Supplementary Figure 5. (Line 179-193)

References

- Sharma, U. *et al.* Biogenesis and function of tRNA fragments during sperm maturation and fertilization in mammals. *Science* **351**, 391-396 (2016).
- Sharma, U. *et al.* Small RNAs Are Trafficked from the Epididymis to Developing Mammalian Sperm. *Developmental cell* **46**, 481-494 e486 (2018).

Su, Z., Kuscu, C., Malik, A., Shibata, E. & Dutta, A. Angiogenin generates specific stress-induced tRNA halves and is not involved in tRF-3-mediated gene silencing. *The Journal of biological chemistry* **294**, 16930-16941 (2019).

- For their microinjection experiments, why did the authors choose to inject the amount of tRNA that is equivalent to 10 sperm (2 ng/ μ l) when only one sperm contributes ~10-30 fg total of RNA of which tRNAs is a subset and the fact that they had previously been fertilised by sperm (who have already contributed a tRNA pool). This doesn't seem biological. Can authors explain why they didn't use a scrambled RNA control, especially when live rates were so much higher in the sham injection (70%, vs 20-40%, sup table 1) in your injection group, suggesting that injection of that amount of RNA is detrimental to embryo development, so I do not agree with their statement on lines 174-177.

Reply: According to reviewer's instructions, we added more microinjection experiments used scrambled RNA as control (Figure 4f-i and Supplementary Figure 6k-t). The results showed that there was no significant difference in live born (% transfer) between scrambled RNA and Inf sperm 30-40 nt RNAs injection group (47% vs 47%, Supplementary Table 6 and 9).

We apologize again for the mistakes in the previous MS that “numbers of injected zygotes (n=50), transferred zygotes (n=50) and live born (% transfer) (35 (70%)) in Control injection group” was wrong. In fact, the number of injected zygotes is 100, the number of transferred zygotes is 100 and live born (% transfer) is 35% in Control injection group, slightly higher than *Ang*^{+/+} Con (25%, compared with Control injection group, $\chi^2 = 2.381$, df = 1, $P > 0.05$) and lower than *Ang*^{+/+} Inf (36%, compared with Control injection group, $\chi^2 = 0.022$, df = 1, $P > 0.05$), *Ang*^{-/-} Con (40%, compared with Control injection group, $\chi^2 = 0.533$, df = 1, $P > 0.05$) and *Ang*^{-/-} Inf (47%, compared with Control injection group, $\chi^2 = 2.976$, df = 1, $P > 0.05$). Thus, there were no significant differences among groups in live born (% transfer) (Supplementary Table 4 and 9).

Thus, we suggest that sperm 30-40 nt RNAs injection did not adversely affect embryo development compared with control injection.

In the present studies, we chose the amount of RNA for zygotic RNA injection according to the previous for zygotic tsRNAs (Chen et al., 2016; Zhang et al., 2018; Sarker et al., 2019) and microRNAs (Wang et al., 2021) injection researches. We agree with the reviewer that the concentration of RNA for zygotic injection was not biological. But we could not give a biological concentration of tsRNAs because there were basic tsRNAs in sperm that came from sperm during fertilization. In fact, the amount of RNAs equals approximately the total RNA of 10 sperm in all groups include *Ang*^{+/+} Inf, *Ang*^{+/+} Con, *Ang*^{-/-} Inf and *Ang*^{-/-} Con. However, the offspring of *Ang*^{+/+} Con, *Ang*^{-/-} Inf and *Ang*^{-/-} Con group exhibited a similar pattern with the control injection (vehicle only) group. These results indicated that the zygotic injection of excessive sperm RNAs was not responsible for the phenotypes. This finding was consistent with the previous

researches showed that zygotic injection of excessive sperm RNAs from normal mice could not induce any phenotypes in the offspring (Zhang et al., 2018; Sarker et al., 2019). Recently, Shi et al. found that transfection of a pool of tsRNAs promoted lineage differentiation in day 6 embryoid bodies and downregulating the mitochondria oxidative phosphorylation and translation/ribosome pathways in embryo (Shi et al., 2021). The phenotypic outcome of injecting sperm 30-40 nt RNA fractions from *Ang^{+/+}* Inf mice or a pool of synthetic tsRNAs could be a combinatorial effect and might relate to their function in cell fate regulation in the early embryo. Nevertheless, it would be interesting to investigate the function of lower tsRNAs amount by zygotic RNA injection. We appreciate the reviewer's suggestion and will include this in our future study.

It must be pointed out that the live born (% transfer) was significantly smaller in the synthetic tsRNAs than in the scrambled RNA injection group (17% vs 47%, $\chi^2 = 10.793$, $df = 1$, $P < 0.01$, Supplementary Table 6 and 9). Recent researches have clarified that transfection of synthetic tsRNAs promoted the differentiation of embryonic stem cells (Krishna et al., 2019) and lineage in embryoid bodies (Shi et al., 2021). Fewer live fetuses per litter in synthetic tsRNAs group was probably due to synthetic tsRNAs-induced alteration in embryonic development and embryonic mortality. To eliminate the effects of litter size on the growth and metabolic parameters, we adjusted the litters to 4 to 5 pups on postnatal day 2 (because there were 4 to 5 pups per litter in synthetic tsRNAs injection group). There was no significant difference in the body weight of males among groups at the time of weaning. By far, we cannot eliminate the effects of lower litter size on offspring's body weight in synthetic tsRNAs group.

References:

- Chen, Q. *et al.* Sperm tsRNAs contribute to intergenerational inheritance of an acquired metabolic disorder. *Science* **351**, 397-400 (2016).
- Sarker, G. *et al.* Maternal overnutrition programs hedonic and metabolic phenotypes across generations through sperm tsRNAs. *Proceedings of the National Academy of Sciences of the United States of America* **116**, 10547-10556 (2019).
- Shi, J. *et al.* PANDORA-seq expands the repertoire of regulatory small RNAs by overcoming RNA modifications. *Nature cell biology* **23**, 424-436 (2021).
- Wang, Y. *et al.* Sperm microRNAs confer depression susceptibility to offspring. *Science advances* **7**, eabd7605 (2021).
- Zhang, Y. *et al.* Dnmt2 mediates intergenerational transmission of paternally acquired metabolic disorders through sperm small non-coding RNAs. *Nature cell biology* **20**, 535-540 (2018).

- The most interesting comparison of tRNAs is between the *Ang^{+/+}* inflammation and *Ang^{-/-}* inflammation group (Sup table 2), yet no comparison between these two. Also not all tRNAs were restored by *Ang^{-/-}* (sup table 2) which suggests that there are other RNase involved in producing tRNA halves and potentially in the inflammatory paternal mediated offspring phenotype. This was actually found in publication from Su et al (reference 29) 'Angiogenin generates specific stress-induced tRNA halves and is not involved in tRF-3-mediated gene

silencing', which found that the majority of stress-induced tRNA halves, except for the 5_ half from tRNAHisGTG and the 3_ half from tRNAAspGTC, are ANG independent, suggesting there are other RNases that produce tRNA halves. They also found that a group of tRNAs (tRNAGly, tRNAGlu, tRNALys, tRNAVal, tRNAHis, tRNAAsp, and tRNAscC) were more responsive to Ang. The authors may want to think about modifying the graphs in Figure 3 to be more representative of those tRNAs that are more responsive to Ang. But also indicate in text that there are likely other mediators.

Reply: As suggested by the reviewer, there are several specific RNases that can cleave tRNAs at the anticodon loop resulting in their fragmentation into tsRNAs (Chen et al., 2021). Our results showed that the abundance of 5'-tsRNA-Gly, 5'-tsRNA-Glu and 5'-tsRNA-Val were increased in inflammatory caput epididymis and decreased by *Ang* deletion but not 5'-tsRNA-Cys. It was in accordance with small RNA seq results that the abundances of tRNA-Gly, tRNA-Glu and tRNA-Val were higher in cauda sperm of *Ang*^{+/+} Inf mice compared with *Ang*^{+/+} Con, *Ang*^{-/-} Inf and *Ang*^{-/-} Con males (Supplementary Figure 5). Interestingly, these results were consistent with the previous research indicating that 5'-tRNA-Gly, 5'-tRNA-Glu and 5'-tRNA-Val were more responsive to Ang, while 5'-tsRNA-Cys was Ang independent (Su et al., 2019). These results suggested that there were other RNases that produce 5'-tsRNAs.

According to reviewer's instructions, we added a Venn diagram in Figure 3 to show that numbers of 5'-tsRNAs with significant upregulated in sperm from inflammatory mice and downregulated in sperm from *Ang* deleted mice (Figure 3j). Also, the information has been incorporated into the Results and Discussion section.

“*Ang* deletion almost completely negated the effect of inflammation on the sperm 5'-tsRNAs that originate from both nucleus- and mitochondria-encoded tRNAs (Figure 3i-m and Supplementary Table 3). Prominent examples of inflammation-induced tsRNAs include 5'-tsRNA-Gly-GCC, 5'-tsRNA-iMet-CAT, and 5'-mt-tsRNA-Val-TAC, whose abundance displayed ~two- to threefold increases in inflammatory mouse sperm and decreased by *Ang* deletion to baseline levels (Figure 3j-m, Supplementary Figure 4 and Supplementary Table 3). Strikingly, several 5'-tsRNAs such as 5'-tsRNA-Cys-GCA, were upregulated in inflammatory mice but expression levels were not reversed by *Ang* deletion (Figure 3n, Supplementary Figure 4 and Supplementary Table 3).” (Line 166-175)

References:

- Chen, Q., Zhang, X., Shi, J., Yan, M. & Zhou, T. Origins and evolving functionalities of tRNA-derived small RNAs. *Trends in Biochemical Sciences* DOI: 10.1016/j.tibs.2021.1005.1001 (2021).
- Su, Z., Kuscu, C., Malik, A., Shibata, E. & Dutta, A. Angiogenin generates specific stress-induced tRNA halves and is not involved in tRF-3-mediated gene silencing. *Journal of Biological Chemistry* **294**, 16930-16941 (2019).

- Not all offspring measure are completed in each of the models. It makes it seem unfinished. For instance;
o RNA transcription of muscular tissue is presented in the very first model (C57/B16 inflammatory model) but not in the other two models, even if they measured the top changing genes.

Reply: According to reviewer's instructions, we measured the top changing genes by RT-PCR. The information has been incorporated into Supplementary Figure 1, 3 and 6.

o Fat phenotypes, running speed etc. not present in the microinjection models.

Reply: According to reviewer's instructions, the information has been added as Supplementary Figure 6c-f and m-p and incorporated into the Results section.

- Statistical analysis = The statistical analysis should contain mixed models to include both the mother and father in which the offspring were generated from (if more than on (is it 'one'?) offspring was sampled per litter – can't tell) as well as litter size if litter sizes were not normalized. This will ensure that what you are seeing is not biased because of one litter etc.

Reply: According to reviewer's instructions, offspring growth rate during the period 3-12 weeks was analyzed using a MIXED procedure for repeated measures (SAS 9.4), with paternal treatment, age, litter size and the interaction between paternal treatment and age entered as fixed factors, and father as a random factor, repeated age (subject = mother / type = un). The information has been incorporated into the Materials and methods section.

One or two male offspring of the average body weight were selected from each litter for GTT, ITT and PTT. Two or three male offspring of the average body weight were selected from each litter for treadmill test, body weight and organ weight measurements. For synthetic tsRNAs injection experiments, to eliminate the effects of litter size on the growth and metabolic parameters, we adjusted the litters to 4 to 5 pups on postnatal day 2 (because there were 4 to 5 pups per litter in synthetic tsRNAs injection group).

Data were analyzed by two-tailed unpaired Student's *t*-test for Figure 1b, f, i, and 2a and Supplementary Figure 1a-e, g, by two-way ANOVA with uncorrected Fisher's LSD for Figure 1c-e, Figure 2f, Figure 4b, f and Supplementary Figure 3d and 6a, k, by one-way ANOVA with uncorrected Fisher's LSD for Figure 2d-e, g-h; 3k-n, 4c-e, g-i and Supplementary Figure 2a-l, 3a-b, e, f-j, 6b-j, l-t. *n* represents the number of mice used in each group, depending on the availability of mice. The number of experiments is detailed in the figure legends. Four independent sets of RNA samples were used for transcriptome sequencing (Figure 1l and Supplementary Figure 1h-i). Three independent sets of sperm RNA samples were used for small RNA sequencing (Figure 3a-n, Supplementary Figure 4).

Minor Comments:

- Add concentration of HCG use for superovulation to methods.

Reply: The information has been incorporated into the Materials and methods section.

Female mice aged 6 weeks were selected as oocyte donors for superovulation, which were performed by intraperitoneal injection with 7.5 IU pregnant mare serum gonadotropin (PMSG), after 48 h intraperitoneal injection with 5 IU human chorionic gonadotrophin (hCG). (Line 477-478)

- Even though you reference a previous paper, you may want to put a table of the founder phenotypes, mating rates, litter sizes from your C57BL/6J inflammatory model and a separate one for your *Ang*^{-/-} models.

Reply: According to reviewer's instructions, the information has been incorporated into the supplementary materials as Supplementary Table 1 and 2.

- The discussion is a little light on, more information on how Ang is modifying the tRNA pool of sperm through the epididymis transit I think is important to speculate how modifications to tRNA sperm pool are causing their effect at fertilization and therefore offspring phenotypes could be expanded.

Reply: According to reviewer's instructions, the information has been incorporated into the Discussion section.

“There are several specific RNases that can cleave tRNAs at the anticodon loop for fragmentation into tsRNAs^{30,45}. Here, we observed that Ang upregulation in the caput epididymis of inflammatory mice coincides with an increased level of several high-abundance tsRNAs including 5'-tsRNA-Gly, 5'-tsRNA-Glu, and 5'-tsRNA-Val. Importantly, the increases of these tsRNAs were reversed by *Ang* deletion. Previous work found that sncRNAs are first synthesized in the caput epididymis and then trafficked to maturing sperm via epididymosomes^{4,10,11}. This requirement for Ang was further supported by comparison of 5'-tsRNA profiles in cauda sperm of *Ang*^{+/+} Inf mice with those of *Ang*^{+/+} Con, *Ang*^{-/-} Inf, and *Ang*^{-/-} Con males. Building on the recent work characterizing sperm RNA-mediated transmission of paternally acquired traits to offspring^{1,2,4,5}, our work identified *Ang* as a genetic factor that is essential in shaping the sperm tsRNAs profile that is responsible for paternal inflammation-induced metabolic disorders in offspring.” (Line 261-272)

- It would benefit the reader if you put the group labels under each figure.

Reply: According to reviewer's instructions, the figures have been revised.

- If required the muscle transcriptional profiles could go as supplementary.

Reply: According to reviewer's instructions, the information has been incorporated into Supplementary Figure 1.

Reviewer comments, further round review -

Reviewer #1 (Remarks to the Author):

My comments were successfully addressed.

Reviewer #2 (Remarks to the Author):

All my previous concerns are addressed in this revised manuscript. I have no further comments.

Reviewer #3 (Remarks to the Author):

I would like to thank the authors for their extensive revisions and additional experimental work that they have added to the manuscript. I believe that it has strengthened their relationship with Ang mediating paternal inflammation through sperm tsRNAs. I think the manuscript should be accepted pending some additional minor changes.

1. I would like to thank the reviewer for the addition of a scrambled RNA control to their experiments. However, I do note that the scrambled control did also seem to have a change in fat mass and fat to muscle mass ratio compared with the injection control. While this is still lower than the tsRNA injection, it's still an interesting finding, which shows that just injection of random RNAs can also modify offspring phenotypes when injected at the PN stage. The authors failed to mention this in their discussion which I think a couple of sentences would suffice.

2. There is also some evidence now that sperm borne RNAs maybe uniquely marked, which synthetic RNAs would not be able to mimic. (Chen Q, Yan M, Cao Z, Li X, Zhang Y, Shi J, Feng GH, Peng H, Zhang X & Zhang Y et al. 2016b Sperm tsRNAs contribute to inter generational inheritance of an acquired metabolic disorder. Science 351) and why phenotype are never fully recapitulated. The authors may want to also add this to their discussion as to why their synthetic tsRNAs couldn't fully recapitulate their in vivo models or even their tRNA pool injections. The discussion again could still be lengthened.

3. Further information about the scrambled RNA control should also be added to the methods. I'm assuming it was injected at the same rate concentration as the tsRNAs 2ng/ul, was it just random RNAs or a specific sub set etc.

Response to the Reviewers ' Comments

REVIEWERS' COMMENTS

Reviewer #1 (Remarks to the Author):

My comments were successfully addressed.

Reply: We thank the reviewer #1 again for his/her helpful and constructive comments.

Reviewer #2 (Remarks to the Author):

All my previous concerns are addressed in this revised manuscript. I have no further comments.

Reply: We thank the reviewer #2 again for his/her helpful and constructive comments.

Reviewer #3 (Remarks to the Author):

I would like to thank the authors for their extensive revisions and additional experimental work that they have added to the manuscript. I believe that it has strengthened there relationship with Ang mediating paternal inflammation through sperm tsRNAs. I think the manuscript should be accepted pending some additional minor changes.

Reply: We thank the reviewer #3 again for his/her helpful and constructive comments.

1. I would like to thank the reviewer for the addition of a scrambled RNA control to their experiments. However, I do note that the scrambled control did also seem to have a change in fat mass and fat to muscle mass ratio compared with the injection control. While this is still lower than the tsRNA injection, its still an interesting finding, which shows that just injection of random RNAs can also modify offspring phenotypes when injected at the PN stage. The authors failed to mention this in their discussion which I think a couple of sentences would suffice.

Reply: Thanks for the suggestions. In response to this reviewer' comment, the information has been incorporated into the Discussion as follows:

It is noted that mice from zygotic scrambled RNAs injection had higher fat mass compared with those derived from natural mating. This alteration may attribute to the injection of random RNA into the male pronuclei of zygotes combined with embryo transfer. (Lines 297-299)

2. There is also some evidence now that sperm borne RNAs maybe uniquely marked, which synthetic RNAs would not be able to mimic. (Chen Q, Yan M, Cao Z, Li X, Zhang Y, Shi J, Feng GH, Peng H, Zhang X & Zhang Y et al. 2016b Sperm tsRNAs contribute to intergenerational inheritance of an acquired metabolic disorder. Science 351) and why phenotype are never fully recapitulated. The authors may want to also add this to their discussion as to why their synthetic tsRNAs couldn't fully recapitulate their in vivo models or even their tRNA pool injections. The discussion again could still be lengthened.

Reply: We agree with the reviewer that sperm-borne RNAs are uniquely marked that synthetic RNAs would not be able to mimic, such as RNA modifications. It is important to note that zygotic injection with synthetic 5'-tsRNAs can partially but not thoroughly resemble paternal inflammation-induced metabolic disorders in offspring. In response to this reviewer' comment, the information has been incorporated into the Discussion as follows:

Chen et al. found that RNA modifications are essential for tsRNAs to exert their intergenerational effects, as non-modified synthetic 5'-tsRNAs did not induce glucose intolerant and insulin resistant in the offspring ¹. In line with their findings, zygotic injection with synthetic 5'-tsRNAs only partially resembled paternal inflammation-induced metabolic disorders in offspring in the present study. The molecular mechanism underlying 5'-tsRNAs actions appears to be more complex. In the present study, non-modified synthetic 5'-tsRNAs induced significantly less glucose intolerant yet similar obesity compared to the sperm RNAs. It is speculated that 5'-tsRNAs may exert their actions via RNA modifications-dependent or independent manners. Further work is needed to elucidate the function of sperm epitranscriptome in association with tsRNAs-mediated intergenerational effects. (Lines 287-297)

3. Further information about the scrambled RNA control should also be added to the methods. I'm assuming it was injected at the same rate concentration as the tsRNAs 2ng/ul, was it just random RNAs or a specific sub set etc.

Reply: According to reviewer's instructions, the sequences of scrambled RNA (a random sequence) has been incorporated into the Materials and methods section and Supplementary Tab. 4.